# Stealthy Backdoor Attack via Confidence-driven Sampling

**Pengfei He**                                                    *hepengf1@msu.edu*
*Department of Computer Science and Engineering, Michigan State University*

**Yue Xing**                                                      *xingyue1@msu.edu*
*Department of Statistics and Probability, Michigan State University*

**Han Xu**                                                        *xuhan2@arizona.edu*
*Department of Electrical and Computer Engineering, University of Arizona*

**Jie Ren**                                                       *renjie3@msu.edu*
*Department of Computer Science and Engineering, Michigan State University*

**Yingqian Cui**                                                  *cuiyingq@msu.edu*
*Department of Computer Science and Engineering, Michigan State University*

**Shenglai Zeng**                                                 *zengshe1@msu.edu*
*Department of Computer Science and Engineering, Michigan State University*

**Jiliang Tang**                                                  *tangjili@msu.edu*
*Department of Computer Science and Engineering, Michigan State University*

**Makoto Yamada**                                                 *makoto.yamada@oist.jp*
*Machine learning and data science (MLDS), Okinawa Institute of Science and Technology*

**Mohammad Sabokrou**                                            *mohammad.sabokrou@oist.jp*
*Machine learning and data science (MLDS), Okinawa Institute of Science and Technology*

**Reviewed on OpenReview:** *https://openreview.net/forum?id=Flh5EXz8dA*

## Abstract

Backdoor attacks facilitate unauthorized control in the testing stage by carefully injecting harmful triggers during the training phase of deep neural networks. Previous works have focused on improving the stealthiness of the trigger while randomly selecting samples to attack. However, we find that random selection harms the stealthiness of the model. In this paper, we identify significant pitfalls of random sampling, which make the attacks more detectable and easier to defend against. To improve the stealthiness of existing attacks, we introduce a method of strategically poisoning samples near the model's decision boundary, aiming to minimally alter the model's behavior (decision boundary) before and after backdooring. Our main insight for detecting boundary samples is exploiting the confidence scores as a metric for being near the decision boundary and selecting those to poison (inject) the attack. The proposed approach makes it significantly harder for defenders to identify the attacks. Our method is versatile and independent of any specific trigger design. We provide theoretical insights and conduct extensive experiments to demonstrate the effectiveness of the proposed method.

## 1 Introduction

While deep neural networks (DNNs) on large datasets and third-party collaborations demonstrate promising performance in various applications, concerns have been raised about potential malicious triggers injected

into the models. These triggers lead to unauthorized manipulation of the model's outputs during testing, causing a "backdoor" attack (Li et al., 2022; Doan et al., 2021a). In particular, attackers can inject triggers into a small portion of training data in a specific manner, then provide either the poisoned training data or backdoored models trained on it to third-party users (Li et al., 2022). In the inference stage, the injected backdoors are activated via triggers, causing triggered inputs to be misclassified as a target label. In the existing literature, many backdoor attack methods have been developed and demonstrate strong attack performance, e.g., BadNets (Gu et al., 2017), WaNet (Nguyen & Tran, 2021), and label-consistent (Turner et al., 2019). These methods can achieve high attack success rates while maintaining a high accuracy on clean data within mainstream DNNs.

An important research direction in backdoor attacks is to enhance the stealthiness of poisoned samples while ensuring their effectiveness simultaneously. While trigger design (e.g., hidden triggers Saha et al., 2020, clean-label (Turner et al., 2019)) has been a primary focus in existing research, recent studies have increasingly explored sampling methods for selecting optimal data points for poisoning and trigger insertion. However, for sampling methods, most existing works (Toneva et al., 2018; Han et al., 2023; Li et al., 2023; Xia et al., 2023; Wu et al., 2023; Zhu et al., 2023) focus on the attacking effectiveness while ignoring the stealthiness of backdoors. Our preliminary study (in Section 4.1) observes that the randomly selected poisoned samples are highly likely to be detected by the defenders. Such a weakness raises a natural question:

*Is there a better sampling strategy to enhance the stealthiness of backdoors?*

To investigate this question, we follow the common understanding to assume that the attackers can access the training data while maybe only allowed to manipulate a part of the training data. For example, the attackers may contribute malicious data to publicly sourced datasets via uploading their own data online (Li et al., 2022). Besides improving the trigger pattern, they also need a sampling strategy to determine the data to update.

To better understand the behavior of the backdoor attacks, in Section 4.1, we investigate the latent space of the backdoored model to take a closer look at the random sampling strategy. We draw two findings from the visualizations in Figure 1. First, most randomly chosen samples are close to the center of their true classes in the latent space. Second, the closer a sample is from its true class on the clean model, the farther it gets from the target class on the backdoored model (Section 4.1). These two observations reveal an important concern about the "stealthiness" of the random sampling strategy, where the randomly sampled data points may be easily detected as outliers. To gain a deeper understanding, we further build a theoretical analysis of SVM in the latent space (Section 4.3) to demonstrate the relation between the random sampling strategy and attack stealthiness. Moreover, our observations suggest an alternative to random sampling—it is better to select samples closer to the decision boundary. Our preliminary studies show that these **boundary samples** can be manipulated to be closer to the clean samples from the target class and can greatly enhance their stealthiness under potential outlier detection (see Figure 1c and 1d).

Inspired by the above observations, we propose a novel method called **confidence-driven boundary sampling** (CBS). Specifically, we identify boundary samples with low confidence scores based on a surrogate model trained on the clean training set. Intuitively, samples with lower confidence scores are closer to the boundary between their own class and the target class in the latent space Karimi et al. (2019) compared to random samples. Therefore, this strategy makes it more challenging to detect attacks. Moreover, our sampling strategy is independent from existing attack approaches, making it exceptionally versatile. It can be easily integrated with various backdoor attacks, offering researchers and practitioners a powerful tool to enhance the stealthiness of backdoor attacks without requiring extensive modifications to their existing methods or frameworks. Extensive experiments combining proposed confidence-based boundary sampling with various backdoor attacks illustrate the advantage of the proposed method over random sampling.

## 2 Related works

### 2.1 Backdoor attacks and defenses

As mentioned in the introduction, backdoor attacks are shown to be a serious threat to DNN. BadNet (Gu et al., 2017) is the first exploration that attaches a small patch to samples to introduce backdoors into a

DNN model. Later, many efforts are put into developing advanced attacks to boost the performance or improve the resistance against potential defenses. Various trigger designs are proposed, including image blending (Chen et al., 2017), image warpping (Nguyen & Tran, 2021), invisible triggers (Li et al., 2020; Saha et al., 2020; Doan et al., 2021b), clean-label attacks (Turner et al., 2019; Saha et al., 2020), sample-specific triggers (Li et al., 2021b; Souri et al., 2022), etc. These attacking methods have demonstrated strong attack performance (Wu et al., 2022). Meanwhile, the study of effective defenses against these attacks also remains active. One popular type of defense detects outliers in the latent space (Tran et al., 2018; Chen et al., 2018; Hayase et al., 2021; Gao et al., 2019; Chen et al., 2018). Other defenses incorporate neuron pruning (Wang et al., 2019), detecting abnormal labels (Li et al., 2021a), model pruning (Liu et al., 2018), fine-tuing (Sha et al., 2022), etc.

## 2.2 Samplings in backdoor attacks

Despite the development of triggers in backdoor attacks, the impact of poisoned sample selection is also attracting more and more attention. Xia et al. (2022) proposed a filtering-and-updating strategy (FUS) to select samples with higher contributions to the injection of backdoors by computing the forgetting event (Toneva et al., 2018) of each sample. For each iteration, poison samples with low forgetting events will be removed, and new samples will be randomly sampled to fill out the poisoned training set. Han et al. (2023); Li et al. (2023); Xia et al. (2023) followed this line and also adopted the forgetting score for sample selection. Wu et al. (2023); Zhu et al. (2023) leverages masks and $l_2$ distance in representation space respectively to improve the effectiveness of the backdoor. Though these works can improve the success rate of backdoor attacks via sample selection, they ignore the backdoor's ability to resist defenses, known as the 'stealthiness' of backdoors. To the best of our knowledge, we are the first to study the stealthiness problem from the sampling perspective.

# 3 Definition and Notation

This section introduces preliminaries about backdoor attacks, including the threat model considered in this paper and a general pipeline that is applicable to many attacks.

## 3.1 Threat model

We follow the commonly used threat model for the backdoor attacks (Gu et al., 2017; Doan et al., 2021b). We assume that the attacker can access the clean training set and modify a proportion of the training data. Then, the victim trains his own models on this data, and the attacker has no knowledge of this training procedure. In a real-world situation, the attacker can access some clean datasets, and modify a proportion of them by inserting triggers. Then they upload the poisoned data to the Internet and victims unknowingly download and use it for training (Gu et al., 2017; Chen et al., 2017). Note that many existing backdoor attacks (Nguyen & Tran, 2021; Turner et al., 2019; Saha et al., 2020) have already adopted this assumption, and our proposed method does not demand additional capabilities from attackers beyond what is already assumed in the context of existing attack scenarios. Furthermore, our method, detailed in Section 4, addresses practical scenarios where attackers are limited to poisoning samples from a specific subset, not the entire dataset, with empirical results in Section 5.5. For example, an attacker might only control their own data and not have access to alter public datasets.

## 3.2 A general pipeline for backdoor attacks

In the following, we introduce a general pipeline, which is applicable to a wide range of backdoor attacks. The pipeline consists of two components.

**(1) Poison sampling**. Let $D_{tr} = \{(x_i, y_i)\}_{i=1}^n$ denote the set of $n$ clean training samples, where $x_i \in \mathcal{X}$ is each individual input sample with $y_i \in \mathcal{Y}$ as the true class. The attacker selects a subset of data $U \subset D_{tr}$, with $p = |U|/|D_{tr}|$ as the poison rate, where the poison rate $p$ is usually small.

**(2) Trigger injection**. Attackers design a strategy $T$ to inject the trigger $t$ into samples selected in the first step. In specific, given a subset of data $U$, attackers generate a poisoned set $T(U)$ as:

$$T(U) = \{(x', y') | x' = G_t(x), y' = S(x, y), \forall (x, y) \in U\} \tag{1}$$

where $G_t : \mathcal{X} \to \mathcal{X}$ is the attacker-specified poisoned image generator with trigger pattern $t$, which satisfies the following constraints: $G_t(x) \neq x$ and $d(G_t(x), x) \leq \epsilon$ where $d$ is some distance function such as $l_2/l_\infty$ distance, and $S$ indicates the attacker-specified target label generator. After training the backdoored model $f(\cdot; \theta^b)$, where $\theta^b$ denote parameters of the backdoored model, on the poisoned set, the injected backdoor will be activated by trigger $t$. For any given clean test set $D_{te}$, the accuracy of $f(\cdot; \theta^b)$ evaluated on trigger-embedded dataset $T(D_{te})$ is referred to as success rate, and attackers also expect to see high accuracy on any clean samples without triggers.

## 4 Method

In this section, we will first analyze the commonly used random sampling method, and then introduce our proposed method as well as some theoretical understandings.

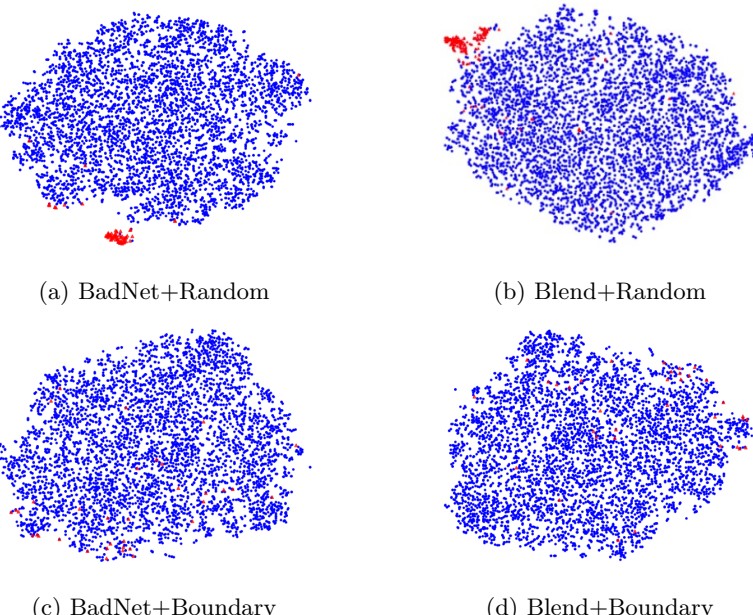

(a) BadNet+Random

(b) Blend+Random

(c) BadNet+Boundary

(d) Blend+Boundary

Figure 1: Latent space visualization of BadNet and Blend via **Random** and **Boundary** sampling.

### 4.1 Revisit random sampling

**Visualization of Stealthiness.** Random sampling selects samples to be poisoned from the clean training set with the same probability and is commonly used in existing attacking methods. However, we suspect that such unconstrained random sampling is easy to detect as outliers of the target class in the latent space. To examine the sample distribution in the latent space, we first conduct TSNE (Van der Maaten & Hinton, 2008) visualizations for the backdoored model of (1) clean samples of the target class, and (2) the poisoned samples from other classes but labeled as the target class. We consider these poisoned samples are obtained by two representative attack algorithms, BadNet (Gu et al., 2017) and Blend (Chen et al., 2017) both of which apply random sampling, on CIFAR10 (Krizhevsky et al., 2009), in Figure 1a and 1b. In detail, the visualizations show the latent representations of samples from the target class, and the colors red and blue indicate poisoned and clean samples respectively. One can see a clear gap between poisoned and clean samples. For both attacks, most of the poisoned samples form a distinct cluster outside the clean

samples. This indicates a separation in latent space which can be easily detected by potential defenses. For example, Spectral Signature (Tran et al., 2018), SPECTRE (Hayase et al., 2021), SCAn (Tang et al., 2021) are representative defenses relying on detecting outliers in the latent space and show great power defending various backdoor attacks (Wu et al., 2022).

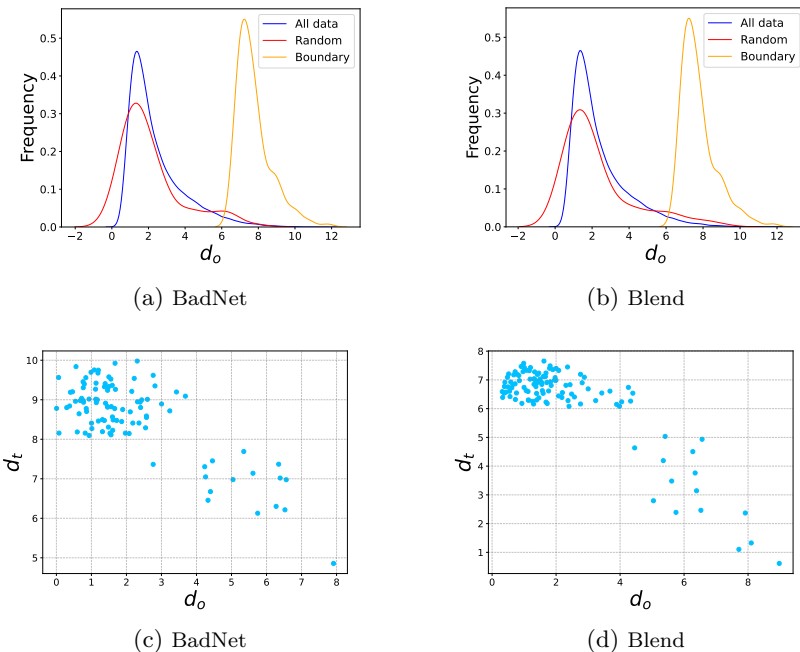

Figure 2: The left two figures depict the distribution of $d_o$ when samples are Randomly selected by BadNet and Blend. The right two figures shows the relationship between $d_o$ and $d_t$ for BadNet and Blend.

**Relation between Stealthiness & Random Sampling.** In our study, we also observe the potential relation between random sampling and the stealthiness of backdoors [1]. To elaborate, we further calculate the distance from each selected sample (without trigger) to the center[2] of their true classes computed on the clean model, which is denoted as $d_o$. As seen in Figure 2a and 2b, random sampling is likely to select samples that are close to the center of their true classes. However, we find $d_o$ may have an obvious correlation with the distance between the sample and the target class which we visualize in the previous Figure 1. Formally, we define the distance between each selected sample (with trigger) and the center of the target class computed on the backdoored model as $d_t$.

From Figure 2c and 2d, we observe a negative correlation between $d_t$ and $d_o$ [3], indicating that samples closer to the center of their true classes in the clean model tend to be farther from the target class after poisoning and thus easier to detect. These findings imply that random sampling often results in the selection of samples with weaker stealthiness. Our observations also suggest that samples closer to the boundary may lead to better stealthiness and motivate our proposed method.

## 4.2 Confidence-driven boundary sampling (CBS)

One key challenge for boundary sampling is how to determine which samples are around the boundaries. Though we can directly compute the distance from each sample to the center of the target class in the latent space and choose those with smaller distances, this approach can be time-consuming, as one needs to compute the center of the target class first and then compute the distance for each sample. This problem can be more severe when the dataset's size and dimensionality grow. Consequently, a more efficient and effective method is in pursuit.

---

[1] We provide a detailed discussion of "stealthiness" in Appendix 8.5.

[2] A formal definition of $d_o$ and $d_t$ is shown in Appendix 8.5

[3] We also include a discussion of the relationship between raw input space and latent space in Appendix 8.11.

To solve this issue, we consider the *confidence score*. To be more specific, we follow the notations from Section 3.2 and further assume there exist $K$ classes, i.e., $\mathcal{Y} = \{1, ..., K\}$, for simplicity. Let $f(\cdot; \theta)$ denote a classifier with model parameter $\theta$, and the output of its last layer is a vector $z \in \mathbb{R}^K$. *Confidence score* is calculated by applying the softmax function on the vector $z$, i.e. $\boldsymbol{s_c(f(x; \theta)) = \sigma(z)} \in [0,1]^K$, where $\sigma(\cdot)$ is the softmax function.

This confidence score is considered the most accessible uncertainty estimate for deep neural network (Pearce et al., 2021) and is shown to be closely related to the decision boundary (Li et al., 2018; Fawzi et al., 2018). Since our primary goal is to identify samples that are closer to the decision boundary, we can find samples with similar confidence for both the true class[4] and the target class. Thus, we can define boundary samples:

**Definition 4.1** (**Confidence-based boundary samples**). Given a data pair $(x, y)$, model $f(\cdot; \theta)$, a confidence threshold $\epsilon$ and a target class $y'$, if

$$|s_c(f(x; \theta))_y - s_c(f(x; \theta))_{y'}| \le \epsilon, \tag{2}$$

then $(x, y)$ is noted as $\epsilon$-boundary sample with target $y'$.

To explain Definition 4.1, since $s_c(f(x; \theta))_y$ represents the probability of classifying $x$ as class $y$, then when there exists another class $y'$, for which $s_c(f(x; \theta))_{y'} \approx s_c(f(x; \theta))_y$, it signifies that the model is uncertain about whether to classify $x$ as class $y$ or class $y'$. This uncertainty suggests that the sample is positioned near the boundary that separates class $y$ from class $y'$ (Karimi et al., 2019).

The proposed **Confidence-driven boundary sampling** (CBS) method is based on Definition 4.1. In general, CBS selects boundary samples in Definition 4.1 for a given threshold $\epsilon$. Since we assume the attacker has no knowledge of the victim's model, we apply a surrogate model like what black-box adversarial attacks often do (Chakraborty et al., 2018). In detail, a pre-trained surrogate model $f(\cdot; \theta)$ is leveraged to estimate confidence scores for each sample, and $\epsilon$-boundary samples with pre-specified target $y^t$ are selected for poisoning. The detailed algorithm is shown in Algorithm 1 [5]. Note that the threshold $\epsilon$ is closely related to poison rate $p$ in Section 3.2, and we can determine $\epsilon$ so that $|U(y^t, \epsilon)| = p \times |\mathcal{D}_{tr}|$. Since we claim that our sampling method can be easily adapted to various backdoor attacks, we provide an example that adapts our sampling methods to Blend (Chen et al., 2017), where we first select samples to be poisoned via Algorithm 1 and then blend these samples with the trigger pattern $t$ to generate the poisoned training set.

---

**Algorithm 1** CBS

---

**Input** Clean training set $\mathcal{D}_{tr} = \{(x_i, y_i)\}_{i=1}^N$, model $f(\cdot; \theta)$, pre-train epochs $E$, threshold $\epsilon$, target class $y^t$
**Output** Poisoned sample set $U$, poisoned label set $S_p$.
  Pre-train the surrogate model $f$ on $\mathcal{D}_{tr}$ for $T$ epochs and obtain $f(\cdot; \theta)$
  Initialize poisoned sample set $U = \{\}$
  **for** $i = 1, ..., N$ **do**
    **if** $|s_c(f(x_i; \theta))_{y_i} - s_c(f(x_i; \theta))_{y^t}| \le \epsilon$ **then**
      $U = U \cup \{(x_i, y_i)\}$
    **end if**
  **end for**
  Return poisoned sample set $U$

---

### 4.3 Theoretical understandings

To better understand CBS, we conduct theoretical analysis on a simple SVM model. As shown in Figure 3, in a 2-dimensional space, we consider a binary classification task where two classes are uniformly distributed in two balls centered at $\mu_1$(orange circle) and $\mu_2$(blue circle) with radius $r$ respectively in latent space[6]:

$$C_1 \sim p_1(x) = \frac{1}{\pi r^2} 1[\|x - \mu_1\|_2 \le r], \text{ and}$$
$$C_2 \sim p_2(x) = \frac{1}{\pi r^2} 1[\|x - \mu_2\|_2 \le r], \tag{3}$$

---

[4]For a correctly classified sample, the true class possesses the largest score.
[5] Discussion of computation overhead is included in Appendix 8.4
[6]This analysis is suitable for any neural networks whose last layer is a fully connected layer.

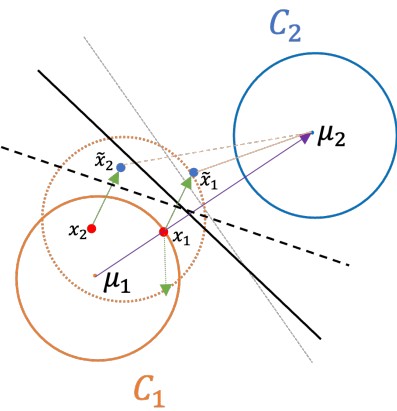

Figure 3: Backdoor on SVM

where let $\mu_2 = 0$ for simplicity. Assume that each class contains $n$ samples. We consider a simple attack that selects one single sample $x$ from class $C_1$, add a trigger to it to generate a poisoned $\tilde{x}$, and assign a label as class $C_2$ for it. Let $\tilde{C}_1, \tilde{C}_2$ denote the poisoned data, and we can obtain a new backdoored decision boundary of SVM on the poisoned data. To study the backdoor effect of the trigger, we assume $\tilde{x} = x + \epsilon t/\|t\|$ where $t/\|t\|, \epsilon$ denote the direction and strength of the trigger, respectively.

To explain this design, we assume that the trigger introduces a 'feature' to the original samples (Khaddaj et al., 2023), and this 'feature' is closely related to the target class while nearly orthogonal to the prediction features[7]. In addition, we assume $t$ is fixed for simplicity, which means this trigger is universal and we argue that this is valid because existing attacks such as BadNet (Gu et al., 2017) and Blend (Chen et al., 2017) inject the same trigger to every sample. To ensure the backdoor effect, we further assume $(\mu_2 - \mu_1)^T t \geq 0$, otherwise the poisoned sample will be even further from the target class (shown as the direction of the green dashed arrow) and lead to subtle backdoor effects. We are interested in two questions: **(Q1)** Are boundary samples harder to detect? **(Q2)** How do samples affect the backdoor performance?

To investigate **(Q1)**, we adopt the Mahalanobis distance (Mahalanobis, 2018) between the poisoned sample $\tilde{x}$ and the target class $\tilde{C}_2$ as an indicator of outliers. A smaller distance means $\tilde{x}$ is less likely to be an outlier, indicating better stealthiness. For **(Q2)**, we estimate the success rate by estimating the volume (or area in 2D data) of the shifted class $C_1$ to the right of the backdoored decision boundary. This is because when triggers are added to every sample, the whole class will shift in the direction of $t$, shown as the orange dashed circle in Figure 3. The following series of theorems and propositions answer the above two questions. We begin with the Manahalnobis distance:

**Theorem 4.2** (Mahalanobis distance). *Assume $\tilde{x} = x + \epsilon t/\|t\|_2 := x + a$ for some arbitrary $x$ and some trigger $t$ and strength $\epsilon$. Also assume $\mu_2 = 0$, $(\mu_2 - \mu_1)^T \tilde{x} \geq 0$. Denote $\tilde{\mu}_2$ and $\tilde{S}_2$ as the sample mean and covariance matrix of the poisoned data with label $C_2$. Then the Mahalanobis distance between $\tilde{x}$ and the target class $\tilde{C}_2$ is defined as*

$$d_M^2(\tilde{x}, \tilde{C}_2) = (\tilde{x} - \tilde{\mu}_2)^T \tilde{S}_2^{-1}(\tilde{x} - \tilde{\mu}_2).$$

*There exists some large constant $n_0$ so that when $n \geq n_0$, $d_M^2(\tilde{x}, \tilde{C}_2)$ satisfies*

$$P\left(\left|d_M^2(\tilde{x}, \tilde{C}_2) - \frac{4\|\tilde{x}\|^2}{r^2}\right| \geq t\right) \leq c_1 \exp\left(-c_2 t^2 n\right) \tag{4}$$

*for some positive constants $c_1$ and $c_2$.*

The proof of Theorem 4.2 can be found in Appendix 8.1. Since $\tilde{\mu}_2$ and $\tilde{S}_2$ are sample mean and covariance, we use concentration inequalities for vector average and matrix average to describe their behavior.

---

[7]Prediction feature here is referred to features used for prediction when no triggers involved.

Theorem 4.2 shows the Mahalanobis distance $(d_M^2(\tilde{x}, \tilde{C}_2))$ from the fixed $\tilde{x}$ to $\tilde{C}_2$. When $n$ increases, $d_M^2(\tilde{x}, \tilde{C}_2)$ converges to its limit $4\|\tilde{x}\|^2/r^2$. In addition, to answer **(Q1)**, the limit $4\|\tilde{x}\|^2/r^2$ is determined by the distance between $\tilde{x}$ and $\mu_2(=0)$: A larger $\|\tilde{x}\|$ results in a larger Mahalanobis distance, leading to a higher chance of $\tilde{x}$ being detected as an outlier (i.e., identified by the defender). To compare CBSand random selecting, if the attacker selects the support vector to attack, then $4\|\tilde{x}\|^2/r^2$ is minimized. Otherwise, $4\|\tilde{x}\|^2/r^2$ will be larger.

In the following, to answer **(Q2)**, we first discuss the attack success rate assuming infinite training samples and then study how the finite sample affects the selection of the poisoned data and the margin of SVM.

**Theorem 4.3** (Attack success rate, population). *Under the same conditions as Theorem 4.2, assume $n \to \infty$ and a hard margin exists, then the success rate for an arbitrary $\tilde{x} = x + a$ is an increasing function of*

$$\epsilon \cos(a, \tilde{x} - \mu_1) - \|\tilde{x} - \mu_1\|/2 - r/2.$$

The proof of Theorem 4.3 can be found in Appendix 8.1. To figure out the attack success rate, we directly calculate the area of incorrect classification.

*Remark* 4.4 (Existence of hard margin). Theorem 4.3 describes how the attack on $x$ affects the attack success rate. However, the existence of a hard margin depends on the selected sample $x$ and trigger $t$, which in turn determines whether an attack is effective. We provide Figure 4 for a better illustration. For the trigger $t$, we need $t^T(-\mu_1) > 0$, which indicates that the trigger $t$ moves the clean sample $x$ (from $C_1$) towards the target cluster $C_2$. To guarantee a hard margin, we require the poisoned sample $\tilde{x}$ to fall into the shaded area, which is determined by $f_1, f_2$ and $C_1$. $f_1, f_2$ are the common tangent lines to $C_1, C_2$, and are two extreme hyperplane that separates two clusters. Formally, we can define the area as $x \in \{x \,|\, (\{\|x + a - \mu_1\|_2 \geq r\} \cap \{f_1(x + a) < 0\} \cap \{f_2(x + a) > 0\}) \cup \{f_1(x + a) > 0\} \cup \{f_2(x + a) < 0\}\}$. These conditions indicate that samples closer to the decision boundary are more likely to result in a hard margin in the poisoned data and guarantee an effective attack.

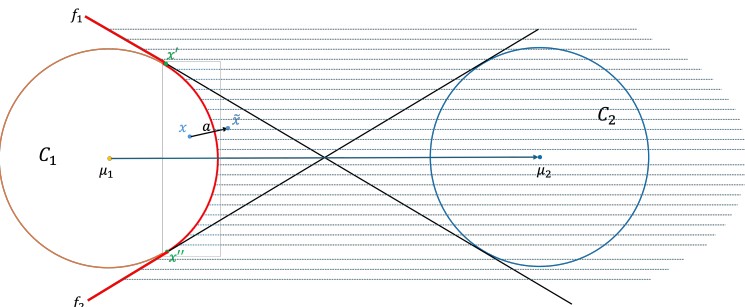

Figure 4: An illustrating figure for the existence of hard margin. The shaded area represents the region of $\tilde{x}$ where a hard margin exists.

On the other hand, unlike random sampling, since CBSrelies on the estimate of the confidence, we provide the following results to illustrate the impact of a finite sample size on CBSand random sampling. We first present Theorem 4.5 below to explain the change of the SVM margin under the finite-sample scenario using clean data:

**Theorem 4.5** (Finite-sample scenario). *Under the same conditions as Theorem 4.2, consider the classification with clean data. Assume there are $n$ samples from $C_1$ and $n$ samples from $C_2$. Take $\delta_n = (\log n)/\sqrt{n}$. With probability at least*

$$1 - 2\left(1 + 2\left\lceil \frac{d_{1,f} - d_{1,\min}}{\delta_n} \right\rceil\right)\left(1 - \frac{c\log^2 n}{n\pi r^2}\right)^n,$$

*uniformly for all hyperplane which separates $C_1$ and $C_2$, the corresponding margin for the $2n$ samples is $O(\sqrt{\delta_n})$-close to the margin for $C_1$ and $C_2$. The terms $d_{1,f}$ and $d_{1,\min}$ are constant values, and their definition can be found in (8) and (7) in Appendix 8.1.*

The proof of Theorem 4.5 is in Appendix 8.1. The general idea is to construct some regions in $C_1$ and $C_2$ along their boundary and demonstrate that with a high probability, there is at least one sample that falls in each of the regions. Then we use these regions to quantify the difference between the margin to the population and the margin to the finite samples.

Based on Theorem 4.5, with a large enough $n$, the margin of SVM converges in probability. While Theorem 4.5 describes the margin using clean training data, the results can be further extended to discuss the sample selected by CBS, as well as the margin under poisoned data. In the following, Proposition 4.6 is for CBS, and Proposition 4.7 is for random sampling.

**Proposition 4.6** (CBS in finite-sample scenario). *Under the conditions in Theorem 4.5, with the same probability as in Theorem 4.5, the sample selected by CBS is $O(\sqrt{\delta_n})$-close to the support vector in the population. If $t$ and $x$ are chosen such that the hard margin exists following Remark 4.4, the margin of the decision boundary determined by the finite poisoned samples is $O(\sqrt{\delta_n})$-close to its population version.*

**Proposition 4.7** (Random poisoning in finite-sample scenario). *Under the conditions in Theorem 4.5, with the same probability as in Theorem 4.5, if $t$ and $x$ are chosen such that the hard margin exists following Remark 4.4, then the margin of the decision boundary determined by the finite poisoned samples is $O(\sqrt{\delta_n})$-close to its population version.*

Proposition 4.6 and 4.7 shows the consistency of these methods: When $n \to \infty$, the sample selected by CBS is close to the support vector in the population version. In addition, the margin of both methods converges to their population version respectively.

*Remark* 4.8 (Effectiveness-stealthiness trade-off). Based on the theorem, a smaller $\|\tilde{x}\|_2$ results in a smaller $d_M^2$, reducing the likelihood of being detected as an outlier. Additionally, closer proximity between $\tilde{x}$ and $\mu_1$ corresponds to a higher success rate without defenses. These observations highlight the trade-off between stealthiness and backdoor performance without defenses. Our experiments in Section 5 further demonstrate that incorporating boundary samples significantly improves stealthiness with only a slight reduction in success rate without defenses.[8]

*Remark* 4.9 (Hard margin does not exist). We also compare cases when the poisoned sample $x$ is too far from the target center $\mu_2$. When the poisoned sample is far enough from $\mu_2$ and the decision boundary, e.g., the poisoned sample $\tilde{x}$, is still within reach of its true class, a hard margin will not exist. In this case, the misclassification of the single poisoned example will be ignored when $n$ is large enough, and the decision boundary of the SVM (with a soft margin) will be the same as the one from the clean SVM. Consequently, the poisoning effect is significantly reduced. To achieve a better success rate, the attacker needs to poison more samples, which can cause inefficiency and worse stealthiness. Therefore, poisoning samples closer to the boundary can even achieve better effectiveness while maintaining stealthiness.

## 5 Experiment

In this section, we conduct experiments to validate the effectiveness of CBS, and show its ability to boost the stealthiness of various existing attacks. We evaluate CBS and baseline samplings under no-defense and various representative defenses in Section 5.2 and 5.3. In particular, we select poisoned samples from the whole training data in these three sections and provide results when only partial data is accessible in Section 5.5 to validate the effectiveness of our approach in a broad and practical scenario. In Section 5.6, we will provide more empirical evidence to illustrate that CBS is harder to detect and mitigate. We also direct readers to additional experiments regarding larger datasets and more defenses in the Supplementary for a more comprehensive evaluation.

### 5.1 Experimental settings

To evaluate CBS and show its ability to be applied to various kinds of attacks, we consider 3 types [9] of attacking methods that cover most of existing backdoor attacks.

---

[8]Code can be found in `https://github.com/PengfeiHePower/boundary-backdoor`.
[9]We determine the types based on the threat models of attacking methods.

Table 1: Performance on Type I backdoor attacks (Cifar10).

| Model Defense | Attacks | ResNet18 | | | ResNet18 $\to$ VGG16 | | |
|---|---|---|---|---|---|---|---|
| | | Random | FUS | CBS | Random | FUS | CBS |
| No Defenses | BadNet | 99.9±0.2 | 99.9±0.1 | 93.6±0.3 | 99.7±0.1 | 99.9±0.06 | 94.5±0.4 |
| | Blend | 89.7±1.6 | 93.1±1.4 | 86.5±0.6 | 81.6±1.3 | 86.2±0.8 | 78.3±0.6 |
| | Adapt-blend | 76.5±1.8 | 78.4±1.2 | 73.6±0.6 | 72.2±1.9 | 74.9±1.1 | 68.6±0.5 |
| | Adapt-patch | 97.5±1.2 | 98.6±0.9 | 95.1±0.8 | 93.1±1.4 | 95.2±0.7 | 91.4±0.6 |
| SS | BadNet | 0.5±0.3 | 4.7±0.2 | **20.2±0.3** | 1.9±0.9 | 3.6±0.6 | **11.8±0.4** |
| | Blend | 43.7±3.4 | 42.6±1.7 | **55.7±0.9** | 16.5±2.3 | 17.4±1.9 | **21.5±0.8** |
| | Adapt-blend | 62±2.9 | 61.5±1.4 | **70.1±0.6** | 38.2±3.1 | 36.1±1.7 | **43.2±0.9** |
| | Adapt-patch | 93.1±2.3 | 92.9±1.1 | **93.7±0.7** | 49.1±2.7 | 48.1±1.3 | **52.9±0.6** |
| STRIP | BadNet | 0.4±0.2 | 8.5±0.9 | **23.7±0.8** | 0.8±0.3 | 9.6±1.5 | **15.7±1.2** |
| | Blend | 54.7±2.7 | 57.2±1.6 | **60.6±0.9** | 49.1±2.3 | 50.6±1.7 | **56.9±0.8** |
| | Adapt-blend | 0.7±0.2 | 5.5±1.8 | **8.6±1.2** | 1.8±0.9 | 3.9±1.1 | **6.3±0.7** |
| | Adapt-patch | 21.3±2.1 | 24.6±1.8 | **29.8±1.2** | 26.5±1.7 | 27.8±1.3 | **29.7±0.5** |
| ABL | BadNet | 16.8±3.1 | 17.3±2.3 | **31.3±1.9** | 14.2±2.3 | 15.7±2.0 | **23.6±1.7** |
| | Blend | 57.2±3.8 | 55.1±2.7 | **65.7±2.1** | 55.1±1.9 | 53.8±1.3 | **56.2±1.1** |
| | Adapt-blend | 4.5±2.7 | 5.1±2.3 | **6.9±1.7** | 25.4±2.6 | 24.7±2.1 | **28.3±1.7** |
| | Adapt-patch | 5.2±2.3 | 7.4±1.5 | **8.7±1.3** | 10.8±2.7 | 11.1±1.5 | **13.9±1.3** |
| NC | BadNet | 1.1±0.7 | 13.5±0.4 | **24.6±0.3** | 2.5±0.9 | 14.4±1.3 | **17.5±0.8** |
| | Blend | 82.5±1.7 | **83.7±1.1** | 81.7±0.6 | **79.7±1.5** | 77.6±1.6 | 78.5±0.9 |
| | Adapt-blend | 72.4±2.3 | 71.5±1.8 | **74.2±1.2** | 59.8±1.7 | 59.2±1.2 | **62.1±0.6** |
| | Adapt-patch | 2.2±0.7 | 6.6±0.5 | **14.3±0.3** | 10.9±2.3 | 13.4±1.4 | **16.2±0.9** |

In detail, **Type I** backdoor attacks allow attackers to inject triggers into a proportion of training data and release the poisoned data to the public. Victims train models on them from scratch. The attack aims to misclassify samples with triggers as the pre-specified target class (also known as the all-to-one scenario). **Type II** backdoor attacks share the same threat model with **Type I** attacks, and the difference is that victims finetune pre-trained models on poisoned data, and the adversary's goal is to misclassify samples from one specific class with triggers as the pre-specified target class (also known as the one-to-one scenario). Distinct from the preceding categories, **Type III** backdoor attacks necessitate an additional degree of control over the training process of the victim's model. This control affords attackers the ability to concurrently optimize both the backdoor triggers and the model parameters, particularly in all-to-one attack scenarios.

**Baselines for sampling**. We compare CBS with two baselines—Random and FUS (Xia et al., 2022). The former selects samples to be poisoned with a uniform distribution, and the latter selects samples that contribute more to the backdoor injection via computing the forgetting events (Toneva et al., 2018) for each sample. In our evaluation, we focus on image classification tasks on datasets Cifar10 and Cifar100 (Krizhevsky et al., 2009)[10], and model architectures ResNet18 (He et al., 2016), VGG16 (Simonyan & Zisserman, 2014). We use ResNet18 as the surrogate model [11] for CBS and FUS if not specified. The surrogate model is trained on the clean training set via SGD for 60 epochs, with an initial learning rate of 0.01 and reduced by 0.1 after 30 and 50 epochs. We implement CBS according to Algorithm.1 and follow the original setting in (Xia et al., 2022) to implement FUS, i.e., 10 overall iterations and 60 epochs for updating the surrogate model in each iteration. After the generation of poisoned samples, we test the attacking performance on ResNet18 (the same architecture as the surrogate model) as well as transferring to another model architecture VGG16 (denoted as ResNet18 $\to$ VGG16 in tables 123).

## 5.2 Performance of CBS in Type I backdoor attacks

**Attacks & Defenses.** We consider 3 representative attacks in this category—BadNet (Gu et al., 2017) which attaches a small patch pattern as the trigger to samples to inject backdoors into neural networks;

---

[10]Additional datasets are included in Appendix 8.7

[11] Discussion of surrogate models in Appendix 8.10

Table 2: Performance on Type II backdoor attacks.

| | Model Defense | Attacks | ResNet18 | | | ResNet18 → VGG16 | | |
|---|---|---|---|---|---|---|---|---|
| | | | **Random** | **FUS** | **CBS** | **Random** | **FUS** | **CBS** |
| **CIFAR10** | **No Defenses** | Hidden-trigger | 81.9±1.5 | 84.2±1.2 | 76.3±0.8 | 83.4±2.1 | 86.2±1.3 | 79.6±0.7 |
| | | LC | 90.3±1.2 | 92.1±0.8 | 87.2±0.5 | 91.7±1.4 | 93.7±0.9 | 87.1±0.8 |
| | **NC** | Hidden-trigger | 6.3±1.4 | 5.9±1.1 | **9.7±0.9** | 10.7±2.4 | 11.2±1.5 | **14.7±0.6** |
| | | LC | 8.9±2.1 | 8.1±1.6 | **12.6±1.1** | 11.3±2.6 | 9.8±1.1 | **12.9±0.9** |
| | **FP** | Hidden-trigger | 11.7±2.6 | 9.9±1.3 | **14.3±0.9** | 8.6±2.4 | 8.1±1.4 | **11.8±0.8** |
| | | LC | 10.3±2.1 | 13.5±1.2 | **20.4±0.7** | 7.9±1.7 | 8.2±1.1 | **10.6±0.7** |
| | **ABL** | Hidden-trigger | 1.7±0.8 | 5.6±1.6 | **10.5±1.1** | 3.6±1.1 | 8.8±0.8 | **10.4±0.6** |
| | | LC | 0.8±0.3 | 8.9±1.5 | **12.1±0.8** | 1.5±0.7 | 9.3±1.2 | **12.6±0.8** |
| **CIFAR100** | **No Defenses** | Hidden-trigger | 80.6±2.1 | 84.1±1.8 | 78.9±1.3 | 78.2±2.3 | 81.4±1.6 | 75.8±1.2 |
| | | LC | 86.3±2.3 | 87.2±1.4 | 84.7±0.9 | 84.7±2.8 | 85.2±1.4 | 81.5±1.1 |
| | **NC** | Hidden-trigger | 3.8±1.4 | 4.2±0.9 | **7.6±0.7** | 4.4±1.1 | 5.1±1.2 | **6.8±0.9** |
| | | LC | 6.1±1.8 | 5.4±1.1 | **8.3±0.5** | 3.9±1.2 | 3.8±0.9 | **8.3±0.7** |
| | **FP** | Hidden-trigger | 15.3±3.1 | 16.7±0.9 | **23.2±0.7** | 8.9±1.3 | 9.3±1.1 | **12.3±0.7** |
| | | LC | 13.8±2.7 | 12.7±1.5 | **16.9±0.6** | 10.3±1.4 | 9.9±0.8 | **14.2±0.5** |
| | **ABL** | Hidden-trigger | 2.3±0.9 | 3.9±1.3 | **6.5±1.1** | 3.7±0.9 | 3.5±0.7 | **6.4±0.4** |
| | | LC | 0.9±0.2 | 2.7±0.8 | **6.2±0.6** | 2.5±0.8 | 2.1±0.7 | **6.7±0.5** |

Blend (Chen et al., 2017) which applies the image blending to interpolate the trigger with samples; and Adaptive backdoor[12] (Qi et al., 2022) which introduces regularization samples to improve the stealthiness of backdoors, as backbone attacks. We include 4 representative defenses: Spectral Signiture (SS) (Tran et al., 2018) and STRIP (Gao et al., 2019) which are outlier-detection-based defenses, Anti-Backdoor Learning (ABL) (Li et al., 2021a) and Neural Cleanser (NC) (Wang et al., 2019) which are not detection-based defenses. We follow the default settings for backbone attacks and defenses. For CBS, we set $\epsilon = 0.2$ and the corresponding poison rate is 0.2% applied for Random and FUS, to guarantee that poisoning rates are the same for all sampling methods. We retrain victim models on poisoned training data from scratch via SGD for 200 epochs with an initial learning rate of 0.1 and decay by 0.1 at epochs 100 and 150. Then we compare the success rate which is defined as the probability of classifying samples with triggers as the target class. We repeat every experiment 5 times and report average success rates (ASR) as well as the standard error if not specified [13]. Results on Cifar10 are shown in Table 1, and results on Cifar100 and Tiny-ImageNet are shown in the Appendix.

**Performance comparison**. Generally, CBS enhances the resilience of backbone attacks against various defense mechanisms. It achieves notable improvement compared to Random and FUS without a significant decrease in ASR when no defenses are in place. This is consistent with our analysis in Section 4.3. We notice that CBS has the lowest success rate when no defenses are active, which is consistent with our analysis in Remark 4.8[14]. Nonetheless, CBS still achieves commendable performance, with success rates exceeding 70% and even reaching 90% for certain attacks. It is important to note that the effectiveness of CBS varies for different attacks and defenses. The improvements are more pronounced when dealing with stronger defenses and more vulnerable attacks. For instance, when facing SS, a robust defense strategy, CBS significantly enhances ASR for nearly all backbone attacks, especially for BadNet. In this case, CBS can achieve more than a 20% increase compared to Random and a 15% increase compared to FUS. Additionally, it's worth mentioning that CBS consistently strengthens resistance against detection-based (first two) and non-detection-based defenses (the other two). This further supports the notion that boundary samples are inherently more challenging to detect and counteract. While the improvement of CBS on VGG16 is slightly less pronounced than on ResNet18, it still outperforms Random and FUS in nearly every experiment, indicating that CBS can be effective even on unknown models.

---

[12]Both Adaptive-Blend and Adaptive-Patch are included

[13]Accuracy on the clean samples are included in Appendix 8.7

[14]Additional discussion can be found in Appendix 8.8

Table 3: Performance on Type III backdoor attacks.

| | Model Defense | Attacks | ResNet18 Random | FUS | CBS | VGG16 Random | FUS | CBS |
|---|---|---|---|---|---|---|---|---|
| **CIFAR10** | No Defenses | Lira | 91.5±1.4 | 92.9±0.7 | 88.2±0.8 | 98.3±0.8 | 99.2±0.5 | 93.6±0.4 |
| | | WaNet | 90.3±1.6 | 91.4±1.3 | 87.9±0.7 | 96.7±1.4 | 97.3±0.9 | 94.5±0.5 |
| | | WB | 88.5±2.1 | 90.9±1.9 | 86.3±1.2 | 94.1±1.1 | 95.7±0.8 | 92.8±0.7 |
| | NC | Lira | 10.3±1.6 | 12.5±1.1 | **16.1±0.7** | 14.9±1.5 | 18.3±1.1 | **19.6±0.8** |
| | | WaNet | 8.9±1.5 | 10.1±1.3 | **13.4±0.9** | 10.5±1.1 | 12.2±0.7 | **13.7±0.9** |
| | | WB | 20.7±2.1 | 19.6±1.2 | **27.2±0.6** | 23.1±1.3 | 24.9±0.8 | **28.7±0.5** |
| | STRIP | Lira | 81.5±3.2 | 82.3±2.3 | **87.7±1.1** | 82.8±2.4 | 81.5±1.7 | **84.6±1.3** |
| | | WaNet | 80.2±3.4 | 79.7±2.5 | **86.5±1.4** | 77.6±3.1 | **79.3±2.2** | 78.2±1.5 |
| | | WB | 80.1±2.9 | 81.7±1.8 | **86.6±1.2** | 83.4±2.7 | 82.6±1.8 | **87.3±1.1** |
| | FP | Lira | 6.7±1.7 | 6.2±1.2 | **12.5±0.7** | 10.4±1.1 | 9.8±0.8 | **13.3±0.6** |
| | | WaNet | 4.8±1.3 | 6.1±0.9 | **8.2±0.8** | 6.8±0.9 | 6.4±0.6 | **8.3±0.4** |
| | | WB | 20.8±2.3 | 21.9±1.7 | **28.3±1.1** | 25.7±1.3 | 26.2±1.2 | **29.1±0.7** |
| **CIFAR100** | No Defenses | Lira | 98.2±0.7 | 99.3±0.2 | 96.1±1.3 | 97.1±0.8 | 99.3±0.4 | 94.5±0.5 |
| | | WaNet | 97.7±0.9 | 99.1±0.4 | 94.3±1.2 | 96.3±1.2 | 98.7±0.9 | 94.1±0.7 |
| | | WB | 95.1±0.6 | 96.4±1.1 | 94.7±0.9 | 93.2±0.9 | 96.7±0.4 | 91.9±0.8 |
| | NC | Lira | 0.2±0.1 | 1.7±1.2 | **5.8±0.9** | 3.4±0.7 | 3.9±1.0 | **7.2±0.9** |
| | | WaNet | 1.6±0.8 | 3.4±1.3 | **8.2±0.8** | 2.9±0.6 | 2.5±0.8 | **5.1±1.2** |
| | | WB | 7.7±1.5 | 7.5±0.9 | **15.7±0.7** | 8.5±1.3 | 7.6±0.9 | **14.9±0.7** |
| | STRIP | Lira | 84.3±2.7 | 83.7±1.5 | **87.2±1.1** | 82.7±2.5 | 83.4±1.8 | **87.8±1.4** |
| | | WaNet | 82.5±2.4 | 82.0±1.6 | **83.9±0.9** | 81.4±2.7 | **84.5±1.7** | 82.6±0.8 |
| | | WB | 85.8±1.9 | 86.4±1.2 | **88.1±0.8** | 82.9±2.4 | 82.3±1.5 | **86.5±1.4** |
| | FP | **Lira** | 7.4±1.9 | 8.9±1.1 | **15.2±0.9** | 8.5±3.2 | 11.8±2.4 | **14.7±1.1** |
| | | **WaNet** | 6.7±1.7 | 6.3±0.9 | **11.3±0.7** | 9.7±2.9 | 9.3±1.8 | **12.6±1.3** |
| | | **WB** | 19.2±1.5 | 19.7±0.7 | **26.1±0.5** | 17.6±2.4 | 18.3±1.7 | **24.9±0.8** |

## 5.3 Performance of CBS in Type II backdoor attacks

**Attacks & Defenses.** We consider 2 representative attacks in this category—Hidden-trigger (Saha et al., 2020), which adds imperceptible perturbations to samples to inject backdoors, and Clean-label (LC) (Turner et al., 2019), which leverages adversarial examples to train a backdoored model. We follow the default settings in the original papers and adapt $l_2$-norm bounded perturbation (perturbation size 6/255) for LC. We test all attacks against three representative defenses that are applicable to these attacks. We include NC, SS, Fine Pruning (FP) (Liu et al., 2018), Anti-Backdoor Learning (ABL) (Li et al., 2021a). We set $\epsilon = 0.3$ for CBS and $p = 0.2\%$ for Random and FUS correspondingly. For every experiment, a source class and a target class are randomly chosen, and poisoned samples are selected from the source class. The success rate is defined as the probability of misclassifying samples from the source class with triggers as the target class. Results on dataset Cifar10 and Cifar100 are presented in Table 2. We include additional results on Tiny-ImageNet in the Supplementary for a further illustration.

**Performance comparison.** As detailed in Table 2, CBS displays an enhanced capacity to withstand various defense mechanisms, akin to Type I attacks, while sacrificing a marginal degree of success rate. Notably, in the presence of defenses, CBS consistently surpasses both Random and FUS strategies in performance, demonstrating its adaptability across various challenging conditions. This is particularly evident when tackling susceptible attack strategies like BadNet, where CBS not only achieves substantial gains—outperforming Random by upwards of 10% and FUS by more than 5%—but also maintains smaller standard errors. These smaller errors reflect CBS's stability, which is critical in real-world applications where consistent performance is crucial.

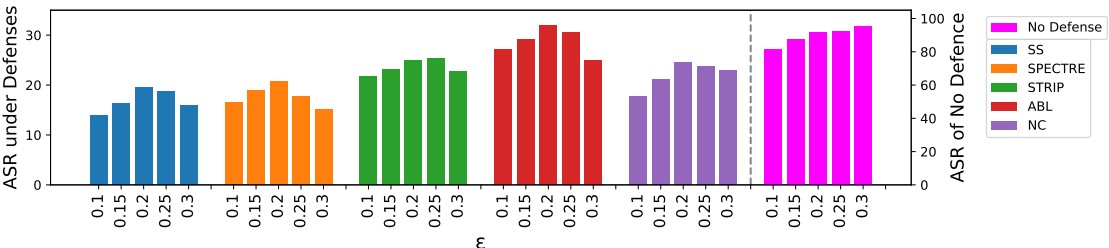

Figure 5: An illustration on the influence of $\epsilon$ in CBS when applied to BadNet. The magenta bar represents ASR without defenses while the left bars present ASR under defenses.

## 5.4 Performance of CBS in Type III backdoor attacks

**Attacks & Defenses.** We consider 3 Representative attacks in this category—Lira (Doan et al., 2021b) which involves a stealthy backdoor transformation function and iteratively updates triggers and model parameters; WaNet (Nguyen & Tran, 2021) which applies the image warping technique to make triggers more stealthy; Wasserstein Backdoor (WB) (Doan et al., 2021a) which directly minimizes the distance between poisoned and clean representations. Note that Type III attacks allow the attackers to take control of the training process. Though our threat model does not require this additional capability of attackers, we follow this assumption when implementing these attacks. Therefore, we directly select samples based on ResNet18 and VGG16 rather than using ResNet18 as a surrogate model. We conduct 3 representative defenses that are applicable for this type of attacks—NC, STRIP, FP. We follow the default settings to implement these attacks and defenses. We set $\epsilon = 0.37$ which matches the poison rate $p = 0.1$ in the original settings of backbone attacks. Results on Cifar10 and Cifar100 are presented in Table 3.

**Performance comparison.** Except for the common findings in previous attacks, where CBS consistently outperforms baseline methods in nearly all experiments, we observe that the impact of CBS varies when applied to different backbone attacks. Specifically, CBS tends to yield the most significant improvements when applied to WB, while its effect is less pronounced when applied to WaNet. For example, when confronting FP and comparing CBS with both Random and FUS, we observed an increase in ASR of over 7% on WB. In comparison, the increase on WaNet amounted to only 3%, with Lira showing intermediate results. This divergence may be attributed to the distinct techniques employed by these attacks to enhance their resistance against defenses. WB focuses on minimizing the distance between poisoned samples and clean samples from the target class in the latent space. By selecting boundary samples that are closer to the target class, WB can reach a smaller loss than that optimized on random samples, resulting in improved resistance. The utilization of the fine-tuning process and additional information from victim models in Lira enable a more precise estimation of decision boundaries and the identification of boundary samples. WaNet introduces Gaussian noise to some randomly selected trigger samples throughout the poisoned dataset, which may destroy the impact of CBS if some boundary samples move away from the boundary after adding noise. These observations suggest that combining CBS with proper trigger designs can achieve even better performance, and it is an interesting topic to optimize trigger designs and sampling methods at the same time for more stealthiness, which leaves for future exploration.

## 5.5 CBS with Partial Backdoor

We investigate scenarios where attackers can only manipulate partial training data. Specifically, we conduct experiments on the ResNet18 model using the Cifar10 dataset, employing the BadNet attack method with various sampling strategies. We designate different subset rates (10%, 5%, 1%) of the training set as accessible to the attacker, who can only poison this fraction of the data. From their accessible data, attackers insert triggers into 10% of the samples. The effectiveness of these attacks, under different defense mechanisms, is evaluated. Our findings, presented in Table 4, demonstrate that our method effectively enhances the stealthiness of backdoor attacks, even with limited data access. This underscores the practical potential of our approach in real-world situations where attackers cannot access the entire training dataset.

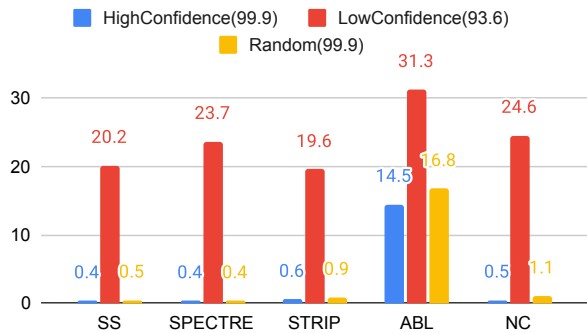

Figure 6: Illustrating impacts of confidence.

Table 4: Experiments for partially poisoned data. Conducted on model ResNet18 and dataset Cifar10, attacking method BadNet is incorporated with different sampling methods.

|  | Subset rate | Random | FUS | CBS |
|---|---|---|---|---|
| **No defenses** | 10% | **99.9** | **99.9** | 93.7 |
|  | 5% | 98.4 | **99.7** | 92.9 |
|  | 1% | 97.2 | **99.7** | 92.3 |
| **SS** | 10% | 1.2 | 6.8 | **15.7** |
|  | 5% | 0.9 | 5.3 | **12.4** |
|  | 1% | 0.6 | 2.8 | **8.5** |
| **NC** | 10% | 2.7 | 8.2 | **14.4** |
|  | 5% | 1.4 | 6.5 | **10.7** |
|  | 1% | 2.2 | 5.3 | **8.4** |
| **Strip** | 10% | 1.5 | 4.9 | **13.2** |
|  | 5% | 0.9 | 3.1 | **9.5** |
|  | 1% | 0.5 | 2.8 | **7.2** |

## 5.6 Ablation study

**Impact of $\epsilon$.** Threshold $\epsilon$ is one key hyperparameter in CBS to determine which samples are around the boundary, and to study the impact of $\epsilon$, we conduct experiments on different $\epsilon$. Since the size of the poisoned set generated by different $\epsilon$ is different, we fix the poison rate to be 0.1% (50 samples), and for large $\epsilon$ that generates more samples, we randomly choose 50 samples from it to form the final poisoned set. We consider $\epsilon = 0.1, 0.15, 0.2, 0.25, 0.3$, and conduct experiments on model ResNet18 and dataset Cifar10 with BadNet as the backbone. Results of ASR under no defense and 5 defenses are shown in Figure 5. It is obvious that the ASR for no defenses is increasing when $\epsilon$ is increasing. We notice that large $\epsilon$ (0.25,0.3) has higher ASR without defenses but relatively small ASR against defenses, indicating that the stealthiness of backdoors is reduced for larger $\epsilon$. For small $\epsilon$ (0.1), ASR decreases for either no defenses or against defenses. These observations suggest that samples too close or too far from the boundary can hurt the effect of CBS, and a proper $\epsilon$ is needed to balance between performance and stealthiness.

**Impact of confidence.** Since our core idea is to select samples with lower confidence, we conduct experiments to compare the influence of high-confidence and low-confidence samples. In detail, we select low-confidence samples with $\epsilon = 0.2$ and high-confidence samples with $\epsilon = 0.9$[15]. We still conduct experiments on ResNet18 and Cifar10 with BadNet, and the ASR is shown in Figure 6. Note that low-confidence samples significantly outperform the other 2 types of samples, while high-confidence samples are even worse than random samples. Therefore, these results further support our claim that low-confidence samples can improve the stealthiness of backdoors.

---

[15]Here we refer to the different direction of Eq.4.1, i.e. $|s_c(f(x;\theta))_y - s_c(f(x;\theta))_{y'}| \geq \epsilon$

We also conduct experiments to study the effect of poisoning rate, and due to the page limit, we include it in Appendix 8.9.

## 6 Conclusion

In this paper, we highlight a crucial aspect of backdoor attacks that was previously overlooked. We find that the choice of which samples to poison plays a significant role in a model's ability to resist defense mechanisms. To address this, we introduce a confidence-driven boundary sampling approach, which involves carefully selecting samples near the decision boundary. This approach has proven highly effective in improving an attacker's resistance against defenses. It also holds promising potential for enhancing the robustness of all backdoored models against defense mechanisms.

## 7 Acknowledgment

This project is supported by MEXT KAKENHI Grant Number 24K03004 and SPS KAKENHI Grant Number 24K20806.

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

## 8 Appendix

### 8.1 Proofs for Section 4.3

Recall the settings in Section 4.3 in the main paper. Suppose two classes $C_1, C_2$ form two uniform distributions of balls centered at $\mu_1, \mu_2$ with radius $r$ in the latent space, i.e.

$$C_1 \sim p_1(x) = \frac{1}{\pi r^2} 1[\|x - \mu_1\|_2 \le r], \text{ and } C_2 \sim p_2(x) = \frac{1}{\pi r^2} 1[\|x - \mu_2\|_2 \le r]$$

Both classes have $n$ samples. Assume $x \in C_1$, and a trigger is added to $x$ such that $\tilde{x} = x + \epsilon t/\|t\|_2 := x + a$. Then define the poisoned data as $\tilde{C}_1 = C_1/\{x\}$ and $\tilde{C}_2 = C_1 \cup \{\tilde{x}\}$. Then we train a backdoored SVM on the poisoned data. The following theorem provides estimations for Mahalanobis distance which serves as the indicator of outliers, and success rate.

*Proof of Theorem 4.2.* Given $n$ samples $x_1, x_2, \ldots, x_n$ from $C_2$ together with the extra example $x$, the Mahalanobis distance becomes

$$d_M^2(\tilde{x}, \tilde{C}_2) = (\tilde{x} - \tilde{\mu}_2)^T \tilde{S}_2^{-1}(\tilde{x} - \tilde{\mu}_2),$$

where $\tilde{\mu}_2$ is the sample mean of the $n$ samples from $C_2$ and the poisoned example $\tilde{x} = x + \epsilon t/\|t\|_2$. The notation $\tilde{S}_2$ denotes the sample covariance matrix.

To show (4), we need to study the behavior of $\tilde{\mu}_2$ and $\tilde{S}_2^{-1}$.

For $\tilde{\mu}_2$, following the vector Bernstein inequality in Kohler & Lucchi (2017), for some constants $c_1$ and $c_2$,

$$P\left(\left\|\frac{1}{n}\sum_{i=1}^n x_i - \mu_2\right\| \ge t\right) \le \exp\left(-c_1 n t^2 + c_2\right).$$

As a result,

$$
\begin{aligned}
&P\left(\|\tilde{\mu}_2 - \mu_2\| \ge t\right) \\
\le\ & P\left(\frac{1}{n+1}\|\tilde{x} - \mu_2\| + \left\|\frac{1}{n+1}\sum_{i=1}^n x_i - \frac{n}{n+1}\mu_2\right\| \ge t\right) \\
\le\ & 1\left(\|\tilde{x} - \mu_2\| \ge \frac{t(n+1)}{2}\right) + P\left(\left\|\frac{1}{n}\sum_{i=1}^n x_i - \mu_2\right\| \ge \frac{n+1}{n}t\right) \\
\le\ & 1\left(\|\tilde{x} - \mu_2\| \ge \frac{t(n+1)}{2}\right) + \exp\left(-c_1 t^2 \frac{(n+1)^2}{n} + c_2\right),
\end{aligned}
$$

where $1(\cdot)$ is the indicator function. In terms of $\tilde{S}_2$, it can be decomposed as

$$
\begin{aligned}
\tilde{S}_2 \ =\ & \frac{1}{n+1}\sum_{i=1}^n (x_i - \tilde{\mu}_2)(x_i - \tilde{\mu}_2)^T + \frac{1}{n+1}(\tilde{x} - \tilde{\mu}_2)(\tilde{x} - \tilde{\mu}_2)^T \\
=\ & \frac{1}{n+1}\sum_{i=1}^n (x_i - \mu_2 + \mu_2 - \tilde{\mu}_2)(x_i - \mu_2 + \mu_2 - \tilde{\mu}_2)^T + \frac{1}{n+1}(\tilde{x} - \tilde{\mu}_2)(\tilde{x} - \tilde{\mu}_2)^T \\
=\ & \frac{1}{n+1}\sum_{i=1}^n (x_i - \mu_2)(x_i - \mu_2)^T + \frac{1}{n+1}\sum_{i=1}^n (\mu_2 - \tilde{\mu}_2)(x_i - \mu_2)^T \\
& + \frac{1}{n+1}\sum_{i=1}^n (\mu_2 - \tilde{\mu}_2)(x_i - \mu_2)^T + \frac{n}{n+1}(\mu_2 - \tilde{\mu}_2)(\mu_2 - \tilde{\mu}_2)^T + \frac{1}{n+1}(\tilde{x} - \tilde{\mu}_2)(\tilde{x} - \tilde{\mu}_2)^T.
\end{aligned}
$$

Denote $\Sigma_2$ as the population covariance matrix of the samples in $C_2$, and $x_i \in \mathbb{R}^d$. Following matrix Bernstein inequality in Tropp et al. (2015), we obtain that for some $c_3$ and $c_4$,

$$P\left(\left\|\frac{1}{n+1}\sum_{i=1}^n (x_i - \mu_2)(x_i - \mu_2)^T - \Sigma_2\right\| \ge t\right) \le 2d \exp\left(-\frac{c_3 t^2 n}{1 + c_4 t}\right).$$

Besides, we also have

$$P\left(\left\|\frac{1}{n+1}\sum_{i=1}^{n}(\mu_2-\tilde{\mu}_2)(x_i-\mu_2)^T\right\|\geq t\right)$$

$$\leq\quad P\left(\|\mu_2-\tilde{\mu}_2\|\left\|\frac{1}{n+1}\sum_{i=1}^{n}(x_i-\mu_2)\right\|\geq t\right)$$

$$\leq\quad P\left(\|\mu_2-\tilde{\mu}_2\|\geq t_1\right)+P\left(\left\|\frac{1}{n+1}\sum_{i=1}^{n}(x_i-\mu_2)\right\|\geq\frac{t}{t_1}\right).$$

As a result,

$$P\left(\|\tilde{S}_2-\Sigma_2\|\geq t\right)$$

$$\leq\quad P\left(\left\|\frac{1}{n+1}\sum_{i=1}^{n}(x_i-\mu_2)(x_i-\mu_2)^T-\Sigma_2\right\|\geq\frac{t}{5}\right)+2P\left(\left\|\frac{1}{n+1}\sum_{i=1}^{n}(\mu_2-\tilde{\mu}_2)(x_i-\mu_2)^T\right\|\geq\frac{t}{5}\right)$$

$$+P\left(\|\mu_2-\tilde{\mu}_2\|^2>\frac{n+1}{n}\frac{t}{5}\right)+1\left(\|\tilde{x}-\tilde{\mu}_2\|^2\geq\frac{(n+1)t}{5}\right)$$

$$\leq\quad 2d\exp\left(-\frac{c_3t^2n/25}{1+c_4t/5}\right)+2P(\|\tilde{\mu}_2-\mu_2\|\geq t_1)+2P\left(\left\|\frac{1}{n+1}\sum_{i=1}^{n}(x_i-\mu_2)\right\|\geq\frac{t}{5t_1}\right)$$

$$+P\left(\|\mu_2-\tilde{\mu}_2\|^2>\frac{n+1}{n}\frac{t}{5}\right)+1\left(\|\tilde{x}-\tilde{\mu}_2\|^2\geq\frac{(n+1)t}{5}\right).$$

In terms of the inverse of $\tilde{S}_2$, following similar steps for (A.6) in Ing & Lai (2011), we also have $\tilde{S}_2^{-1}-\Sigma_2^{-1}=\Sigma_2^{-1}(\Sigma_2-\tilde{S}_2)\tilde{S}_2^{-1}$, thus

$$\begin{aligned}P\left(\left\|\tilde{S}_2^{-1}-\Sigma_2^{-1}\right\|\geq t\right)&=&P\left(\left\|\Sigma_2^{-1}(\Sigma_2-\tilde{S}_2)\tilde{S}_2^{-1}\right\|\geq t\right)\\&\leq&P\left(\left\|\Sigma_2^{-1}\right\|\left\|\Sigma_2-\tilde{S}_2\right\|\left\|\tilde{S}_2^{-1}\right\|\geq t\right)\\&=&P\left(\left\|\Sigma_2^{-1}\right\|\left\|\Sigma_2-\tilde{S}_2\right\|\geq t\left\|\tilde{S}_2\right\|\right)\\&\leq&P\left(\left\|\Sigma_2^{-1}\right\|\left\|\Sigma_2-\tilde{S}_2\right\|\geq t(\left\|\Sigma_2\right\|-\left\|\Sigma_2-\tilde{S}_2\right\|)\right)\\&=&P\left(\left\|\Sigma_2-\tilde{S}_2\right\|\geq\frac{t\|\Sigma_2\|}{\|\Sigma_2^{-1}\|+t}\right).\end{aligned}$$

Given the above probability bounds for $\tilde{\mu}_2$ and $\tilde{S}_2^{-1}$, we can further bound the Mahalanobis distance as

$$P\left(\left|(\tilde{x}-\tilde{\mu}_2)^T\tilde{S}_2^{-1}(\tilde{x}-\tilde{\mu}_2)-\frac{4\|\tilde{x}\|^2}{r^2}\right|\geq t\right)$$

$$\leq\quad P\left(\left|(\tilde{x}-\mu_2+\mu_2-\tilde{\mu}_2)^T(\tilde{S}_2^{-1}-\Sigma_2^{-1}+\Sigma_2^{-1})(\tilde{x}-\mu_2+\mu_2+\tilde{\mu}_2)-\frac{4\|\tilde{x}\|^2}{r^2}\right|\geq t\right)$$

$$\leq\quad P\left(\underbrace{\left|(\tilde{x}-\mu_2)^T\Sigma_2^{-1}(\tilde{x}-\mu_2)-\frac{4\|\tilde{x}\|^2}{r^2}\right|}_{=0}+|(\tilde{x}-\tilde{\mu}_2)^T(\tilde{S}_2^{-1}-\Sigma_2^{-1})(\tilde{x}-\tilde{\mu}_2)|\right.$$

$$\left.+2|(\tilde{x}-\mu_2)^T\Sigma_2^{-1}(\tilde{\mu}_2-\mu_2)|+|(\tilde{\mu}_2-\mu_2)\Sigma_2^{-1}(\tilde{\mu}_2-\mu_2)|\geq t\right)$$

$$\leq\quad P\left(\|\tilde{x}-\tilde{\mu}_2\|^2\|\tilde{S}_2^{-1}-\Sigma_2^{-1}\|\geq\frac{t}{3}\right)+P\left(\|\tilde{x}-\mu_2\|\|\Sigma_2^{-1}\|\|\tilde{\mu}_2-\mu_2\|\geq\frac{t}{3}\right)$$

$$+P\left(\|\tilde{\mu}_2-\mu_2\|^2\|\Sigma_2^{-1}\|\geq\frac{t}{3}\right).$$

Since the poisoned sample $\tilde{x}$ is from $C_1$, and both $C_1$ and $C_2$ are bounded, there exists some constant $c_5$ so that $\|\tilde{x}-\mu_2\|<c_5$ and $\|\tilde{x}-\tilde{\mu}_2\|<5$ uniformly for all possible choice of $\tilde{x}$ and uniformly for all choices of

$\{x_i\}_{i=1,\ldots,n}$. As a result,

$$
P\left(\left|(\tilde{x}-\tilde{\mu}_2)^T \tilde{S}_2^{-1}(\tilde{x}-\tilde{\mu}_2) - \frac{4\|\tilde{x}\|^2}{r^2}\right| \geq t\right)
$$

$$
\leq \quad P\left(\|\tilde{S}_2^{-1}-\Sigma_2^{-1}\| \geq \frac{t}{3c_5^2}\right) + P\left(\|\Sigma_2^{-1}\|\|\tilde{\mu}_2-\mu_2\| \geq \frac{t}{3c_5}\right) + P\left(\|\tilde{\mu}_2-\mu_2\|^2\|\Sigma_2^{-1}\| \geq \frac{t}{3}\right)
$$

$$
= \quad P\left(\|\tilde{S}_2^{-1}-\Sigma_2^{-1}\| \geq \frac{t}{3c_5^2}\right) + P\left(\|\tilde{\mu}_2-\mu_2\| \geq \frac{tr^2}{12c_5}\right) + P\left(\|\tilde{\mu}_2-\mu_2\|^2 \geq \frac{tr^2}{12}\right)
$$

$$
\leq \quad 2d\exp\left(-\frac{c_3 t^2 n \|\Sigma_2\|^2}{25(3c_5^2\|\Sigma_2^{-1}\|+t)^2 + 5c_4 t\|\sigma_2\|(3c_5^2\|\Sigma_2^{-1}\|+t)}\right)
$$

$$
+ \quad 2\left[\mathbb{1}\left(\|\tilde{x}-\mu_2\| \geq \frac{t_1(n+1)}{2}\right) + \exp\left(-c_1 t_1^2 \frac{(n+1)^2}{n} + c_2\right)\right] + 2\exp\left(-c_1\frac{(n+1)^2}{n}+c_2\right)
$$

$$
+ \quad \mathbb{1}\left(\|\tilde{x}-\mu_2\| \geq \frac{n+1}{2}\sqrt{\frac{(n+1)t}{5n}}\right) + \exp\left(-c_1\frac{t}{5}\frac{(n+1)^3}{n^2}+c_2\right)
$$

$$
+ \quad \mathbb{1}\left(\|\tilde{x}-\mu_2\| \geq \frac{n+1}{2}\frac{tr^2}{12c_5}\right) + \exp\left(-c_1\frac{t^2 r^4}{144c_5^2}\frac{(n+1)^2}{n}+c_2\right)
$$

$$
+ \quad \mathbb{1}\left(\|\tilde{x}-\mu_2\| \geq \frac{n+1}{2}\sqrt{\frac{tr^2}{12}}\right) + \exp\left(-c_1\frac{tr^2}{12}\frac{(n+1)^2}{n}+c_2\right).
$$

When $t$ is small while $tn$ is large enough, the indicator functions all become 0. To simplify the above result, there exists some constants $c_6$, $c_7$ so that when $n \geq n_0$ for some large constant $n_0$,

$$
P\left(\left|(\tilde{x}-\tilde{\mu}_2)^T \tilde{S}_2^{-1}(\tilde{x}-\tilde{\mu}_2) - \frac{4\|\tilde{x}\|^2}{r^2}\right| \geq t\right) \leq c_6\exp\left(-c_7 t^2 n\right).
$$

$\square$

*Proof of Theorem 4.3.* To simplify the analysis, we first investigate the scenario of $n \to \infty$, and then discuss the impact of a finite $n$.

As shown in Fig.7a (for Random) and 7b (for CBS), for a given sample $x$ (red point), $\tilde{x} = x + \epsilon\frac{t}{\|t\|_2} := x + a$ (blue point), where $\mu_1^T t \geq 0$. Since $C_2$ is not changed, the backdoored decision boundary (the bold black line) is determined by $\tilde{x}$ and $C_1$. Specifically, the decision boundary is determined by $\tilde{x}$ and center $\mu_1$ Connect the center of $C_1$ with $\tilde{x}$ and we obtain an interaction point on $C_1$, which is $\mu_1 + r\frac{\tilde{x}-\mu_1}{\|\tilde{x}-\mu_1\|_2}$ and the center between it and $\tilde{x}$ is

$$
\tilde{c}_1 = \frac{\mu_1 + r\frac{\tilde{x}-\mu_1}{\|\tilde{x}-\mu_1\|_2} + \tilde{x}}{2}. \tag{5}
$$

Then we can derive the equation for the backdoored decision boundary:

$$
(x - \tilde{c}_1)^T(\tilde{x} - \mu_1) = 0 \tag{6}
$$

where we assume this decision boundary is not overlapped with $C_2$. During the inference, triggers will be added to samples in $C_1$, which means that the circle of $C_1$ will shift by $\epsilon\frac{t}{\|t\|_2}$ (denoted as $\bar{C}_1$) as shown in Fig.7a and 7b, then the yellow area will be misclassified as $C_2$. Thus the success rate without any defenses is determined by the area of the yellow area. Since the circle of $C_1$ is fixed, we only need to compare the distance from the center of $\bar{C}_1$ to the backdoored decision boundary, which is the bold green line in Fig.7a and 7b. Notice that $\mu_1 - \tilde{x}$ is orthogonal to the decision boundary defined in Eq.6, thus the length of the green bold line is the length of $\tilde{\tilde{c}}_1 - \tilde{c}_1$ in the direction of $\mu_1 - \tilde{x}$ where $\tilde{\tilde{c}}_1$ is the center of $\tilde{\bar{C}}_1$, thus the distance

is computed as:

$$d_D(\tilde{x}) = \frac{(\tilde{\tilde{c}}_1 - \tilde{c}_1)^T(\mu_1 - \tilde{x})}{\|\mu_1 - \tilde{x}\|_2} = \frac{1}{\|\tilde{x} - \mu_1\|}\left(\mu_1 + a - \frac{\mu_1 + r\frac{\tilde{x}-\mu_1}{\|\tilde{x}-\mu_1\|} + \tilde{x}}{2}\right)^T(\tilde{x} - \mu_1)$$

$$= \frac{a^T(\tilde{x} - \mu_1)}{\|\tilde{x} - \mu_1\|} - \frac{\|\tilde{x} - \mu_1\|}{2} - \frac{r}{2}$$

$$= \|a\|\cos(a, \tilde{x} - \mu_1) - \frac{\|\tilde{x} - \mu_1\|}{2} - \frac{r}{2}.$$

The above formulation indicates that smaller $\|\tilde{x} - \mu_1\|$ and $\cos(a, \tilde{x} - \mu_1)$ leads to larger $d_D(\tilde{x})$. Therefore, the closer the selected sample to the decision boundary, the smaller the area of the yellow area is, and the smaller the success rate for the backdoored attack. Note that here we only consider the case that $a^T(\tilde{x} - \mu_1) \geq 0$ otherwise the poisoned sample will remain in the original $C_1$. These results reveal the trade-off between stealthiness and performance.

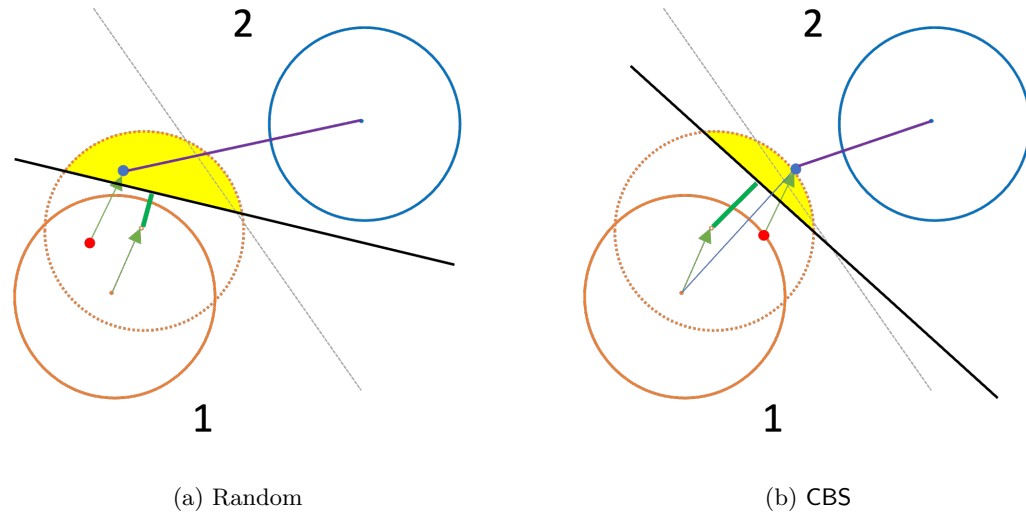

(a) Random  (b) CBS

Figure 7: Illustrating figures for SVM under Random and CBS. The red point is a sample $x$ from $C_1$, and the blue one is the triggered sample $\tilde{x}$. The grey dashed line and black bold line represent the decision boundary of clean and backdoored SVM respectively. We are interested in the Mahalanobis distance between $\tilde{x}$ and the target class $\tilde{C}_2$. The yellow area is in proportion to the success rate and the length of the green bold line is positively correlated with the area of the yellow part. It is obvious that CBShas smaller Mahalanobis distance and smaller area of yellow.

□

*Proof of Theorem 4.5.* Denote $\mathcal{F}$ as the set of all linear functions $f$ which can separate $C_1$ and $C_2$, and define $x^*$ as the point of tangency of the common tangent line to two clusters and $x^* \in C_1$, as shown in Figure 8.

Denote $\delta_n = (\log n)/\sqrt{n}$ and

$$d_{1,\min} = \min_{x \in C_1} \mu_1^T(x - \mu_1), \tag{7}$$

$$d_{1,f} = \mu_1^T(x^* - \mu_1), \tag{8}$$

$d_{2,\max} = \max_{x \in C_2} \mu_1^T(x - \mu_1)$. Then one can define the following regions for $k = 1, \ldots, \lceil(d_{1,f} + \Delta - d_{1,\min})/\delta_n\rceil$ for some small positive constant $\Delta$:

- $C_{1,0} = \{x \in C_1 || \mu_1^T(x - \mu_1)| \in [d_{1,\min}, d_{1,\min} + \delta_n)\}$.

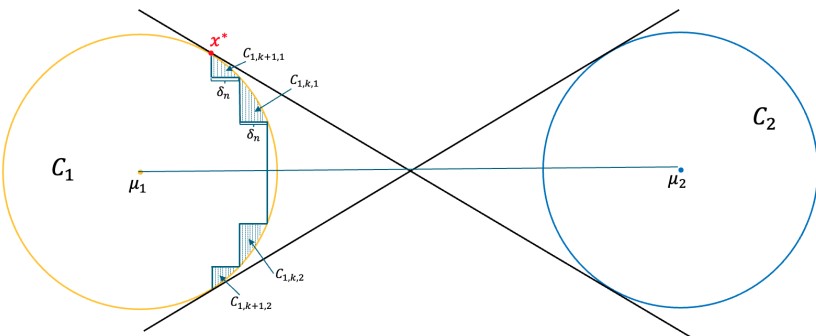

Figure 8: Illustration of impact of finite training samples.

- $C_{1,k,1} = \{x \in C_1 | |\mu_1^T(x - \mu_1)| \in [d_{1,\min} + k\delta_n, d_{1,\min} + (k + 1)\delta_n), \quad \sin(x, \mu_1) > \sup_{x' \in C_{1,k-1,1}} \sin(x', \mu_1)\}$. $C_{1,0,1} = C_{1,0,2} = C_{1,0}$.

- $C_{1,k,2} = \{x \in C_1 | |\mu_1^T(x - \mu_1)| \in [d_{1,\min} + k\delta_n, d_{1,\min} + (k + 1)\delta_n), \quad \sin(x, \mu_1) \leq \inf_{x' \in C_{1,k-1,2}} \sin(x', \mu_1)\}$.

- $C_{2,0} = \{x \in C_2 | |\mu_1^T(x - \mu_1)| \in (d_{2,\max} - \delta_n, d_{2,\max}]\}$. $C_{2,0,1} = C_{2,0,2} = C_{2,0}$.

- $C_{2,k,1} = \{x \in C_2 | |\mu_1^T(x - \mu_1)| \in (d_{2,\max} - (k + 1)\delta_n, d_{2,\max} - k\delta_n], \sin(x, \mu_1) > \sup_{x' \in C_{2,k-1,2}} \sin(x', \mu_1)\}$.

- $C_{2,k,2} = \{x \in C_2 | |\mu_1^T(x - \mu_1)| \in (d_{2,\max} - (k + 1)\delta_n, d_{2,\max} - k\delta_n], \sin(x, \mu_1) \leq \inf_{x' \in C_{2,k-1,2}} \sin(x', \mu_1)\}$.

Note that all the above sets have no overlap with each other, and Figure .

Denote $|C|$ as the area of a region $C$, then it is easy to see that for all $C \in \{C_{i,0}, C_{i,j,k}\}$, $|C| = \Omega((\log n)^2/n)$. Suppose there are $n$ samples in $C_1$, then since all examples are i.i.d. sampled uniformly from $C_1$, we have for some constant $c$, for all $C \in \{C_{i,0}, , C_{i,j,k}\}$,

$$P\left(\forall i, \ x_i \notin C\right) = \left(1 - \frac{|C|}{\pi r^2}\right)^n \leq \left(1 - \frac{c \log^2 n}{n \pi r^2}\right)^n.$$

As a result, for both $i = 1, 2$,

$$P\left(\text{There exists at least one sample in each } C_{i,0} \text{ and } C_{i,k,j} \text{ for } k = 1, \ldots, \left\lceil \frac{d_{1,f} - d_{1,\min}}{\delta_n} \right\rceil, j = 1, 2\right)$$

$$\geq 1 - \left(1 + 2\left\lceil \frac{d_{1,f} - d_{1,\min}}{\delta_n} \right\rceil\right)\left(1 - \frac{c \log^2 n}{n \pi r^2}\right)^n.$$

Based on the above, when taking $n \to \infty$, all the $C_{i,k,j}$ and $C_{i,0}$ regions have at least one sample falls in each of them.

Given the above result, now we look at the minimum distance from the samples in $C_1$ and $C_2$ to any $f \in \mathcal{F}$. For a region $C$, denote $d(C) = \sup_{x_1, x_2 \in C} \|x_1 - x_2\|$. Then for $C_{i,0}$, one can calculate that

$$d(C_{i,2}) = \sqrt{[r^2 - (r - \delta_n)^2] + \delta_n^2} = \sqrt{2r\delta_n}.$$

For the other $C_{i,k,j}$, one can also see that $d(C_{i,k,j}) = O(\sqrt{\delta_n})$.

As a result, if denote $d(f, C)$ as the distance from $f$ to a region $C$, and $d(f, \{x_i\}_{i \in \mathcal{I}})$, then when all the $C_{i,k,j}$ and $C_{i,0}$ regions have at least one sample falls in each of them, we have uniformly for all $f \in \mathcal{F}$,

$$d(f, C_1) = \inf_{k,j} d(f, C_{1,k,j}) = d(f, \{x_i\}_{x_i \in \cup C_{1,k,j}}) + O(\sqrt{\delta_n}) = d(f, \{x_i\}_{i=1,\ldots,n}) + O(\sqrt{\delta_n}),$$

which also implies that the sample selected by CBS is $O(\sqrt{\delta_n})$-close to the point in $C_1$ which is closest to $C_2$. And when $n \to \infty$, $\delta_n \to 0$. $\qquad\square$

## 8.2 Implementation details

In this section, we provide details of attacks and defenses used in experiments as well as implementation details.

### 8.2.1 Implementations for samplings

We implement Random with a uniform distribution on $\mathcal{D}_{tr}$. We implement CBS according to Algorithm 1 in the main paper, and the surrogate model is trained via SGD for 60 epochs with an initial learning rate of 0.01 and decreases by 0.1 at epochs 30,50. We implement FUS according to its original settings, i.e. 10 overall iterations and 60 epochs for updating the surrogate model in each iteration, and the surrogate model is pre-trained the same as in CBS.

### 8.2.2 Attacks

We will provide brief introduction and implementation details for all the backbone attacks implemented in this work.

Type I attacks:

**BadNet (Gu et al., 2017)**. BadNet is the first work exploring the backdoor attacks, and it attaches a small patch to the sample to create the poisoned training set. Then this training set is used to train a backdoor model. We implement it based on the code of work (Qi et al., 2022) and following the default setting.

**Blend (Chen et al., 2017)**. Blend incorporates the image blending technique, and blends the selected image with a pre-specified trigger pattern that has the same size as the original image. We implement this attack based on the code of work (Qi et al., 2022), and following the default setting, i.e. mixing ratio $\alpha = 0.2$.

**Adaptive backdoor (Qi et al., 2022)**. This method leverages regularization samples to weaken the relationship between triggers and the target label and achieve better stealthiness. We implement two versions of the method: Adaptive-blend and Adaptive-patch. During the implementation, we consider the conservatism ratio of $\eta = 0.5$ and mixing ratio $\alpha = 0.2$ for adaptive-blend; conservatism ratio $\eta = 2/3$ and 4 patches for Adaptive-patch.

Type II attacks:

**Hidden-trigger (Saha et al., 2020)**. This attacking method first attaches the trigger to a sample and then searches for an imperceptible perturbation that achieves a similar model output (measured by $l_2$ norm) as the triggered sample. We follow the original settings in work (Saha et al., 2020), i.e. placing the trigger at the right corner of the image, setting the budget size as 16/255, optimizing the perturbation for 10000 iterations with a learning rate of 0.01 and decay by 0.95 for every 2000 iterations.

**Label-consistent (LC) (Turner et al., 2019)**. This attacking method leverages GAN or adversarial examples to create the poisoned image without changing the label. We implement the one with adversarial examples bounded by $l_2$ norm. We set the budget size as 600 to achieve a higher success rate.

Type III attacks:

**Lira (Doan et al., 2021b)**. This method iteratively learns the model parameters and a trigger generator. Once the trigger generator is trained, attackers will finetune the model on poisoned samples attached with triggers generated by the generator, and release the backdoored model to the public. Our implementation is based on the Benchmark (Wu et al., 2022).

**WaNet (Nguyen & Tran, 2021)**. WaNet incorporates the image warping technique to inject invisible triggers into the selected image. To improve the poisoning effect, they introduce a special training mode

that add Gaussian noise to the warping field to improve the success rate. Our implementation is based on the Benchmark (Wu et al., 2022).

**Wasserstein Backdoor (WB) (Doan et al., 2021a)**. This method directly minimizes the distance between poisoned samples and clean samples in the latent space. We follow the original settings, i.e. training 50 epochs for Stage I and 450 epochs for Stage II, set the threshold of constraint as 0.01.

### 8.2.3 Defenses

**Outlier detection defenses:**

**Spectral Signature (SS) (Tran et al., 2018)**. This defense detects poisoned samples with stronger spectral signatures in the learned representations. We remove $1.5 * p$ of samples in each class.

**Activation Clustering (AC) (Chen et al., 2018)**. This defense is based on the clustering of activations of the last hidden neural network layer, for which clean samples and poisoned samples form distinct clusters. We remove clusters with sizes smaller than 35% for each class.

**SCAn (Tang et al., 2021)**. This defense leverages an EM algorithm to decompose an image into its identity part and variation part, and a detection score is constructed by analyzing the distribution of the variation.

**SPECTRE (Hayase et al., 2021)**. This method proposes a novel defense algorithm using robust covariance estimation to amplify the spectral signature of corrupted data. We also remove $1.5 * p$ of samples in each class.

**STRIP (Gao et al., 2019)**. STRIP is a sanitation-based method relying on the observation that poisoned samples are easier to be perturbed, and detect poisoned samples through adversarial perturbations.

**Other defenses:**

**Fine Pruning (FP) (Liu et al., 2018)**. This is a model-pruning-based backdoor defense that eliminates a model's backdoor by pruning these dormant neurons until a certain clean accuracy drops.

**Neural Cleanse (NC) (Wang et al., 2019)**. This is a trigger-inversion method that restores triggers by optimizing the input domain. It is based on the intuition that the norm of reversed triggers from poisoned samples will be much smaller than clean samples.

**Anti-Backdoor Learning (ABL) (Li et al., 2021a)**. This defense utilizes local gradient ascent to isolate 1% suspected training samples with the smallest losses and leverage unlearning techniques to train a cleansed model on poisoned data.

### 8.3 Algorithms

In this section, we provide detailed algorithms for CBS and its application on Blend (Chen et al., 2017).

As shown in Algorithm 1 in the main paper, CBS first pretrain a surrogate model $f(\cdot; \theta)$ on the clean training set $\mathcal{D}_{tr}$ for $E$ epochs; then $f(\cdot; \theta)$ is used to estimate the confidence score for every sample; for a given target $y^t$, samples satisfying $|s_c(f(x_i; \theta))_{y_i} - s_c(f(x_i; \theta))_{y^t}| \leq \epsilon$ are selected as the poison sample set $U$.

As shown in Algorithm 2, the poison sample set $U$ is first selected via Algorithm 1; then for each sample in $U$, a trigger is blended to this sample with a mixing ratio $\alpha$ via $x' = \alpha * t + (1 - \alpha) * x$ and generate the poisoned training set $D_p$.

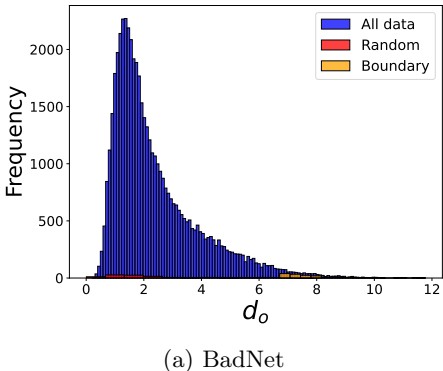

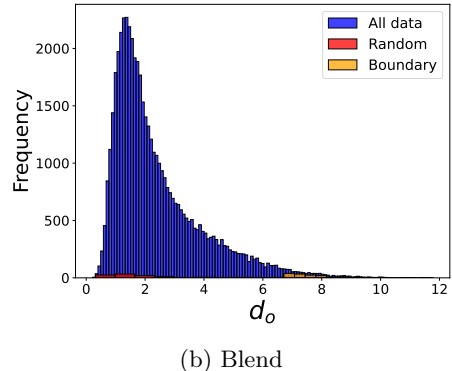

(a) BadNet  (b) Blend

---

**Algorithm 2** Blend+CBS

---

**Input** Clean training set $\mathcal{D}_{tr} = \{(x_i, y_i)\}_{i=1}^N$, surrogate model $f(\cdot; \theta)$, pre-train epochs $E$, threshold $\epsilon$, target class $y^t$, mixing ratio $\alpha$, trigger pattern $t$

**Output** Poisoned training set $D_p$

    Initialize poisoned training set $D_p$

    Select poison set $U$ from $\mathcal{D}_t r$ via Algorithm.1

    **for** $x \in U$ **do**

        Inject triggers to samples: $x' = \alpha * t + (1 - \alpha) * x$

        $D_p = D_p \cup \{x'\}$

    **end for**

    Return poisoned training set $D_p$

---

### 8.4 Discussion of computation overhead of CBS

Since CBS selects samples based on confidence score of a clean model, it is necessary to analyze its computation overhead compared with simple random selection. Suppose the sample size is $N$, poison rate is $r$. For the random selection, we need to randomly select $rN$ samples from the dataset and the time complexity is $O(rN)$. For the proposed method, we need to first compute the confidence score for each sample and then select those smaller than $\epsilon$. Suppose the average time for computing the confidence score for one sample is $t$, and the time complexity for the proposed method is $O(N(t+1))$. Therefore, both complexities are in the linear order of sample size $N$. When the inference time is shorter, CBS is more efficient.

### 8.5 Details for the discussion in Section 4.1

We provide more details for the discussions in Section 4.1.

**Additional figures.** To supplement Figure 2, we present the distributions of distance $d_o$ for different selections of samples. In specific, in Figure 2, we present the kernel density curve of each category for a clearer comparison, and this is why there are curves in the negative region. In the real data of distances, all values are non-negative. The original histograms can be found in Figure 9a and Figure 9b.

**Formal definition of $d_o$ and $d_t$.** We also provide the formal definition of distances $d_o$ and $d_t$ for clearer understanding. Assume sample $x$ comes from the class $y$ and the target class is $y_t$, model $f$ mapping from input space to the latent space, classifier $g$ mapping from latent space to label space. We then can define the center of samples in class $y$ and $y_t$ in the latent space as: $f(x)_{y,mean} := \frac{1}{|\{(x',y) \in D_{tr}\}|} \sum_{(x',y) \in D_{tr}} f(x')$, $\widetilde{f}(x)_{y_t,mean} := \frac{1}{|\{(x',y_t) \in D_p\}|} \sum_{(x',y) \in D_{tr}} \widetilde{f}(x')$, where $f, \widetilde{f}$ denote the clean model and backdoored model respectively. Then we can define $d_o, d_t$ as:

$$d_o(x) := \|f(x) - f(x)_{y,mean}\|_2, \; d_t(x) := \|\widetilde{f}(x+t) - \widetilde{f}(x)_{y_t,mean}\|_2$$

**Additional discussion of stealthiness.** In this work, we focus on the "stealthiness" that poisoned samples are separable in the latent space given the labels. Assume the backdoored model $f$ mapping from input space to the latent representation space, classifier $g$ with label space $Y$, sample $x$ and trigger $t$, and also assume the true label for $x$ as $y$, the triggered label is $y_t$, i.e. $g(f(x)) = y$ and $g(f(x + t)) = y_t$ where $x + t$ denote the combination of input $x$ and trigger $t$. Then the stealthiness is measured by the distance between poisoned sample $x + t$ and the center of clean samples from the target class $y_t$ in the latent space, i.e. $d(f(x+t), f(x)_{y,mean})$ where $f(x)_{y,mean} := \frac{1}{|\{(x',y)\in D_{tr}\}|}\sum_{(x',y)\in D_{tr}} f(x')$. A larger distance means the poisoned sample is separated from the target class and therefore easy to detect, and vice versa.

In the proposed method, since we take the model output before the last linear layer as the latent representation (for example ResNet18), selection based on the confidence score is equivalent to that directly based on the latent representation. To provide some more details, we visualize two classes in Cifar10. In the Figures 10a 10b, there are two clusters with different colors. The blue one is the target class, the green one is the original class, and the red points are the poisoned samples. These verify our statements. In the random selection case, since the poisoned samples are far away from the blue cluster, with the label as the blue class, they are more likely to be identified by the defender. For boundary selection, the poisoned samples are less likely to be detected.

Besides the formal definition above, we would like to clarify that from a more general perspective, the "stealthiness" of an attack is to what extent the attack can be defended. Since defenses are based on different insights, for instance, Spectral Signature (SS) is based on outlier-detection while anti-backdoor (ABL) is based on the fact that poisoned sample is learned faster than clean data, it is hard to find a formal definition for "stealthiness" accommodating all defenses. In our experiments, we test different defenses and the reduction of success rate measure the "stealthiness" (smaller reduction means better resistance against defenses thus more stealthiness).

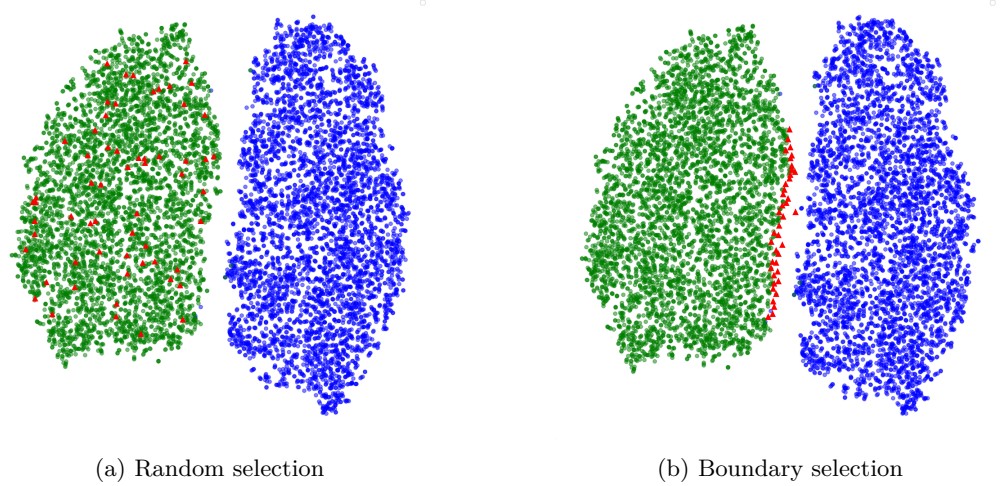

(a) Random selection            (b) Boundary selection

## 8.6 Type III Backdoor Attacks

Due to the page limit of the main text, we present details and comprehensive experimental results in this section.

**Attacks & Defenses.** We consider 3 Representative attacks in this category—Lira (Doan et al., 2021b) which involves a stealthy backdoor transformation function and iteratively updates triggers and model parameters; WaNet (Nguyen & Tran, 2021) which applies the image warping technique to make triggers more stealthy; Wasserstein Backdoor (WB) (Doan et al., 2021a) which directly minimizes the distance between poisoned and clean representations. Note that Type III attacks allow the attackers to take control of the training process. Though our threat model does not require this additional capability of attackers, we follow this assumption when implementing these attacks. Therefore, we directly select samples based on ResNet18 and VGG16 rather than using ResNet18 as a surrogate model. We conduct 5 representative defenses that

are applicable for this type of attacks—SS, NC, STRIP, FP, Activation Clustering (AC) (Chen et al., 2018), including both detection-based (SS,STRIP, AC) and non-detection-based (NC, FP). We follow the default settings to implement these attacks and defenses (details in Appendix 8.2). We set $\epsilon = 0.37$ which matches the poison rate $p = 0.1$ in the original settings of backbone attacks. Results on Cifar10 and Cifar100 are presented in Table 5.

**Performance comparison.** Except for the common findings in previous attacks, where CBS consistently outperforms baseline methods in nearly all experiments, we observe that the impact of CBS varies when applied to different backbone attacks. Specifically, CBS tends to yield the most significant improvements when applied to WB, while its effect is less pronounced when applied to WaNet. For example, when confronting FP and comparing CBS with both Random and FUS, we observed an increase in ASR of over 7% on WB, while the increase on WaNet amounted to only 3%, with Lira showing intermediate results. This divergence may be attributed to the distinct techniques employed by these attacks to enhance their resistance against defenses. WB focuses on minimizing the distance between poisoned samples and clean samples from the target class in the latent space. By selecting boundary samples that are closer to the target class, WB can reach a smaller loss than that optimized on random samples, resulting in improved resistance. The utilization of the fine-tuning process and additional information from victim models in Lira enable a more precise estimation of decision boundaries and the identification of boundary samples. WaNet introduces Gaussian noise to some randomly selected trigger samples throughout the poisoned dataset, which may destroy the impact of CBS if some boundary samples move away from the boundary after adding noise. These observations suggest that combining CBS with proper trigger designs can achieve even better performance, and it is an interesting topic to optimize trigger designs and sampling methods at the same time for more stealthiness, which leaves for future exploration.

## 8.7 Additional experiments

In this section, we provide additional experimental results.

**Type I attacks.** We include additional defenses: Activation Clustering (AC) (Chen et al., 2018), SCAn (Tang et al., 2021), SPECTRE (Hayase et al., 2021), Fine Pruning (FP) (Liu et al., 2018). We also conduct experiments on Cifar100. Results of Type I attacks on Cifar10 and Cifar100 datasets are shown in Table 6 and 7 respectively. CBS has similar behavior on Cifar100—improve the resistance against various defenses while slightly decrease ASR without defenses.

**Type II attacks.** We also include additional defenses: Spectral Signature (SS) (Tran et al., 2018). The results of all defenses on model ResNet18, VGG16 and datasets Cifar10, Cifar100 are presented in Table 8. Detailed analysis is shown in Section 5.3 in the main paper.

**Clean accuracy.** We also report the accuracy of clean samples (without triggers) on the backdoored models of all three sampling methods. Results are shown in Table 9. It is clear that CBS can reach a better clean accuracy than other selections, which makes it more stealthy. To intuitively explain this, since the poisoned samples are around the original decision boundary, they do not severely change the decision boundary, thus preserving a high clean accuracy when no attack is injected in the testing data.

**Tiny-ImageNet dataset**. Except for Cifar10 and Cifar100, we also conduct experiments on a larger dataset Tiny-ImageNet (Le & Yang, 2015). We also train model ResNet18 for 60 epochs to select the poisoned samples for each sampling method. Results are shown in Table 10. These results demonstrate that CBSis applicable to larger datasets and consistently improves the stealthiness of different attacks.

**ImageNet-1k dataset.** We also consider a large-scale dataset, ImageNet-1k (Russakovsky et al., 2015), which has 1000 object classes and more than 1,200,000 images. We test representative backbone attacks from all three types and representative defenses. Since FUS is time-consuming and not eligible for large datasets, we only compare CBS with the random baseline. Results are shown in Table 11. CBS is still effective even on such a big dataset.

Table 5: Performance on Type III backdoor attacks.

| | Model Defense | Attacks | ResNet18 | | | VGG16 | | |
|---|---|---|---|---|---|---|---|---|
| | | | Random | FUS | CBS | Random | FUS | CBS |
| CIFAR10 | No Defenses | Lira | 91.5±1.4 | 92.9±0.7 | 88.2±0.8 | 98.3±0.8 | 99.2±0.5 | 93.6±0.4 |
| | | WaNet | 90.3±1.6 | 91.4±1.3 | 87.9±0.7 | 96.7±1.4 | 97.3±0.9 | 94.5±0.5 |
| | | WB | 88.5±2.1 | 90.9±1.9 | 86.3±1.2 | 94.1±1.1 | 95.7±0.8 | 92.8±0.7 |
| | AC | Lira | 90.7±2.1 | 90.8±1.4 | **91.1±0.9** | 90.5±3.1 | 89.8±2.3 | **91.2±1.2** |
| | | WaNet | **90.5±1.3** | 89.6±0.9 | 89.9±0.6 | 90.8±3.5 | **91.5±2.1** | 90.4±1.4 |
| | | WB | 87.1±2.3 | 87.7±1.5 | **88.2±1.3** | 90.4±2.8 | 89.5±1.7 | **91.1±0.9** |
| | SS | Lira | 86.5±2.7 | 89.6±1.6 | **90.1±1.3** | 90.5±2.5 | **91.3±1.6** | 90.1±1.1 |
| | | WaNet | 87.4±3.1 | **89.4±1.5** | 88.2±1.4 | 90.6±2.6 | 90.8±1.1 | **91.2±0.7** |
| | | WB | 86.4±2.8 | 86.1±2.3 | **88.1±1.7** | 87.6±3.2 | 88.2±2.5 | **89.9±1.3** |
| | NC | Lira | 10.3±1.6 | 12.5±1.1 | **16.1±0.7** | 14.9±1.5 | 18.3±1.1 | **19.6±0.8** |
| | | WaNet | 8.9±1.5 | 10.1±1.3 | **13.4±0.9** | 10.5±1.1 | 12.2±0.7 | **13.7±0.9** |
| | | WB | 20.7±2.1 | 19.6±1.2 | **27.2±0.6** | 23.1±1.3 | 24.9±0.8 | **28.7±0.5** |
| | STRIP | Lira | 81.5±3.2 | 82.3±2.3 | **87.7±1.1** | 82.8±2.4 | 81.5±1.7 | **84.6±1.3** |
| | | WaNet | 80.2±3.4 | 79.7±2.5 | **86.5±1.4** | 77.6±3.1 | **79.3±2.2** | 78.2±1.5 |
| | | WB | 80.1±2.9 | 81.7±1.8 | **86.6±1.2** | 83.4±2.7 | 82.6±1.8 | **87.3±1.1** |
| | FP | Lira | 6.7±1.7 | 6.2±1.2 | **12.5±0.7** | 10.4±1.1 | 9.8±0.8 | **13.3±0.6** |
| | | WaNet | 4.8±1.3 | 6.1±0.9 | **8.2±0.8** | 6.8±0.9 | 6.4±0.6 | **8.3±0.4** |
| | | WB | 20.8±2.3 | 21.9±1.7 | **28.3±1.1** | 25.7±1.3 | 26.2±1.2 | **29.1±0.7** |
| CIFAR100 | No Defenses | Lira | 98.2±0.7 | 99.3±0.2 | 96.1±1.3 | 97.1±0.8 | 99.3±0.4 | 94.5±0.5 |
| | | WaNet | 97.7±0.9 | 99.1±0.4 | 94.3±1.2 | 96.3±1.2 | 98.7±0.9 | 94.1±0.7 |
| | | WB | 95.1±0.6 | 96.4±1.1 | 94.7±0.9 | 93.2±0.9 | 96.7±0.4 | 91.9±0.8 |
| | AC | Lira | 83.5±2.6 | 82.4±1.9 | **87.1±1.3** | 85.2±2.8 | **85.7±2.1** | 84.2±1.2 |
| | | WaNet | 82.7±2.8 | 82.1±2.1 | **86.3±0.9** | 83.8±3.1 | 84.2±1.8 | **85.1±0.9** |
| | | WB | 83.2±2.4 | 84.9±1.6 | **90.2±1.2** | 90.5±2.4 | 89.3±1.5 | **91.8±0.9** |
| | SS | Lira | 93.2±1.7 | **94.6±1.3** | 92.8±0.8 | 91.8±1.9 | 90.7±1.3 | **92.1±0.7** |
| | | WaNet | 92.4±1.9 | **93.3±1.0** | 92.7±0.6 | **90.5±2.3** | 90.1±1.4 | 90.3±1.1 |
| | | WB | 92.9±1.3 | 92.7±0.8 | **94.1±0.9** | 90.1±2.1 | 90.4±1.6 | **92.5±0.8** |
| | NC | Lira | 0.2±0.1 | 1.7±1.2 | **5.8±0.9** | 3.4±0.7 | 3.9±1.0 | **5.2±0.9** |
| | | WaNet | 1.6±0.8 | 3.4±1.3 | **5.2±0.8** | 2.9±0.6 | 2.5±0.8 | **4.1±1.2** |
| | | WB | 7.7±1.5 | 7.5±0.9 | **13.7±0.7** | 8.5±1.3 | 7.6±0.9 | **11.9±0.7** |
| | STRIP | Lira | 84.3±2.7 | 83.7±1.5 | **87.2±1.1** | 82.7±2.5 | 83.4±1.8 | **83.8±1.4** |
| | | WaNet | 82.5±2.4 | 82±1.6 | **83.9±0.9** | 81.4±2.7 | **82.5±1.7** | 82.0±0.8 |
| | | WB | 85.8±1.9 | 86.4±1.2 | **88.1±0.8** | 82.9±2.4 | 82.3±1.5 | **84.5±1.4** |
| | FP | Lira | 87.4±1.9 | 88.2±1.1 | **89.9±0.9** | 82.5±3.2 | 81.8±2.4 | **86.7±1.1** |
| | | WaNet | 86.7±1.7 | 86.3±0.9 | **89.3±0.7** | 81.7±2.9 | 82.1±1.8 | **85.6±1.3** |
| | | WB | 89.2±1.5 | 89.7±0.7 | **92.1±0.5** | 83.6±2.4 | 83.3±1.7 | **87.9±0.8** |

**GTSRB dataset.** GTSRB is a traffic sign recognition benchmark, consisting of traffic signs, and is a different real-world scenario than the standard Cifar10 dataset. We also test representative backbone attacks from all three types and representative defenses. Results are shown in Table 11.

## 8.8 Additional discussion on effectiveness-stealthiness tradeoff

As discussed in Theorem 4.2 and Remark 4.8, there exists an effectiveness-stealthiness tradeoff for attacks. In general, when selecting samples closer to the decision boundary, it is harder to detect but will sacrifice some poisoning effect when facing no defenses. This is also verified by our empirical results in Table 1, Table 2 and Table 3, where CBS has the worst performance when there is no defense.

Table 6: Full Performance on Type I backdoor attacks (Cifar10).

| Model Defense | Attacks | ResNet18 | | | ResNet18 → VGG16 | | |
|---|---|---|---|---|---|---|---|
| | | Random | FUS | CBS | Random | FUS | CBS |
| No Defenses | BadNet | 99.9±0.2 | 99.9±0.1 | 93.6±0.3 | 99.7±0.1 | 99.9±0.06 | 94.5±0.4 |
| | Blend | 89.7±1.6 | 93.1±1.4 | 86.5±0.6 | 81.6±1.3 | 86.2±0.8 | 78.3±0.6 |
| | Adapt-blend | 76.5±1.8 | 78.4±1.2 | 73.6±0.6 | 72.2±1.9 | 74.9±1.1 | 68.6±0.5 |
| | Adapt-patch | 97.5±1.2 | 98.6±0.9 | 95.1±0.8 | 93.1±1.4 | 95.2±0.7 | 91.4±0.6 |
| SS | BadNet | 0.5±0.3 | 4.7±0.2 | **23.2±0.3** | 1.9±0.9 | 3.6±0.6 | **11.8±0.4** |
| | Blend | 43.7±3.4 | 42.6±1.7 | **55.7±0.9** | 16.5±2.3 | 17.4±1.9 | **21.5±0.8** |
| | Adapt-blend | 62±2.9 | 61.5±1.4 | **70.1±0.6** | 38.2±3.1 | 36.1±1.7 | **43.2±0.9** |
| | Adapt-patch | 93.1±2.3 | 92.9±1.1 | **93.7±0.7** | 49.1±2.7 | 48.1±1.3 | **52.9±0.6** |
| AC | BadNet | 0.6±0.3 | 14.2±0.9 | **20.5±0.7** | 5.7±1.2 | 5.3±1.3 | **10.5±1.5** |
| | Blend | 77.1±2.8 | **79.6±2.6** | 77.8±1.4 | 83.1±3.5 | **83.2±2.4** | 81.4±2.1 |
| | Adapt-blend | 76.8±2.1 | 76.1±1.4 | **79.3±1.6** | 69.9±2.8 | 70.6±1.5 | **73.1±1.2** |
| | Adapt-patch | **97.5±2.6** | 94.2±1.7 | 96.6±0.9 | 92.4±2.7 | **93.2±1.4** | 91.3±1.3 |
| SCAn | BadNet | 0.7±0.4 | 10.7±1.2 | **23.5±0.8** | 12.4±1.5 | 10.7±1.2 | **26.4±1.1** |
| | Blend | **84.4±3.4** | 83.6±2.5 | 78.3±2.6 | 80.6±3.2 | **82.1±2.4** | 78.2±0.9 |
| | Adapt-blend | 78.2±2.6 | 77.5±2.1 | **81.5±1.4** | 71.9±2.5 | 71.1±2.1 | **74.4±1.3** |
| | Adapt-patch | **97.5±0.9** | 94.1±0.8 | 96.9±0.4 | 93.1±1.1 | **93.8±0.9** | 91.5±0.5 |
| STRIP | BadNet | 0.4±0.2 | 8.5±0.9 | **26.2±0.8** | 0.8±0.3 | 9.6±1.5 | **15.7±1.2** |
| | Blend | 54.7±2.7 | 57.2±1.6 | **60.6±0.9** | 49.1±2.3 | 50.6±1.7 | **56.9±0.8** |
| | Adapt-blend | 0.7±0.2 | 5.5±1.8 | **8.6±1.2** | 1.8±0.9 | 3.9±1.1 | **6.3±0.7** |
| | Adapt-patch | 21.3±2.1 | 24.6±1.8 | **29.8±1.2** | 26.5±1.7 | 27.8±1.3 | **29.7±0.5** |
| SPECTRE | BadNet | 0.9±0.5 | 10.1±1.4 | **19.6±1.3** | 0.7±0.3 | 8.7±1.2 | **14.9±0.8** |
| | Blend | 9.2±2.4 | 16.7±2.1 | **24.2±1.7** | 8.7±2.6 | 12.8±1.9 | **18.6±0.9** |
| | Adapt-blend | 69±3.5 | 66.8±2.7 | **70.3±1.8** | 67.9±3.2 | 65.2±1.8 | **69.4±0.9** |
| | Adapt-patch | 91.4±1.4 | 89.4±1.2 | **93.1±0.7** | 92.5±2.4 | 91.8±1.4 | **92.1±1.2** |
| ABL | BadNet | 16.8±3.1 | 17.3±2.3 | **31.3±1.9** | 14.2±2.3 | 15.7±2.0 | **23.6±1.7** |
| | Blend | 57.2±3.8 | 55.1±2.7 | **65.7±2.1** | 55.1±1.9 | 53.8±1.3 | **56.2±1.1** |
| | Adapt-blend | 4.5±2.7 | 5.1±2.3 | **6.9±1.7** | 25.4±2.6 | 24.7±2.1 | **28.3±1.7** |
| | Adapt-patch | 5.2±2.3 | 7.4±1.5 | **8.7±1.3** | 10.8±2.7 | 11.1±1.5 | **13.9±1.3** |
| FP | BadNet | 75.2±3.2 | 80.8±2.4 | **81.2±1.3** | 68.3±3.1 | 70.5±2.3 | **73.7±1.1** |
| | Blend | 79.5±3.7 | **81.5±2.4** | 80.4±1.5 | 70.2±2.9 | 72.5±2.1 | **79.3±1.5** |
| | Adapt-blend | **77.5±2.7** | 75.3±2.3 | 77.4±1.2 | 65.1±3.4 | 64.2±2.7 | **68.5±1.6** |
| | Adapt-patch | **97.5±1.1** | 92.7±2.3 | 96.3±0.9 | 93.4±2.2 | 93.3±1.7 | **93.7±0.8** |
| NC | BadNet | 1.1±0.7 | 13.5±0.4 | **24.6±0.3** | 2.5±0.9 | 14.4±1.3 | **17.5±0.8** |
| | Blend | 82.5±1.7 | **83.7±1.1** | 81.7±0.6 | **79.7±1.5** | 77.6±1.6 | 78.5±0.9 |
| | Adapt-blend | 72.4±2.3 | 71.5±1.8 | **74.2±1.2** | 59.8±1.7 | 59.2±1.2 | **62.1±0.6** |
| | Adapt-patch | 2.2±0.7 | 6.6±0.5 | **14.3±0.3** | 10.9±2.3 | 13.4±1.4 | **16.2±0.9** |

To mitigate this limitation, we design two strategies. The general idea is to balance the effectiveness-stealthiness trade-off to control the performance change of different defense methods. In the first strategy, we set a lower threshold for confidence score during selection, i.e. $\epsilon_1 \le |s_c(f(x_i;\theta))_{y_i} - s_c(f(x_i;\theta))_{y_t}| \le \epsilon_2$. The second strategy is to mix the boundary samples with some random samples. We conduct experiments to test if these strategies can improve the performance on undefended models. We test with ResNet18 model, Cifar10 dataset, and backbone attacks are BadNet and Blended. For the first strategy, we set the threshold as $[0.07, 0.25]$ (about 100 samples within this interval and a good balance between effectiveness and stealthiness), and for the second strategy, we select 70% boundary samples and 30% random samples. The poison rate is 0.2%, i.e. 100 poisoned samples. We report the results of both undefended and 4 representative defenses in Table 12.

It is clear that both strategies can improve the attacking performance under no defenses and can achieve comparable performance with random selections. The performance under defenses is slightly reduced because the poisoned samples are easier to detect by defense methods. Nonetheless, the performance still significantly outperforms the baselines.

Table 7: Full Performance on Type I backdoor attacks (Cifar100).

| Model Defenses | Attacks | ResNet18 | | | ResNet18 → VGG16 | | |
|---|---|---|---|---|---|---|---|
| | | Radnom | FUS | Boundary | Radnom | FUS | Boundary |
| No defense | BadNet | 82.8±2.3 | 84.1±1.5 | 78.1±0.9 | 83.1±2.6 | 86.3±1.9 | 80.4±1.2 |
| | Blend | 82.7±2.6 | 83.9±1.7 | 77.9±1.1 | 79.6±2.8 | 82.9±2.1 | 75.2±1.3 |
| | Adapt-blend | 67.1±1.9 | 69.2±1.3 | 64.5±0.7 | 70.6±2.4 | 74.1±1.5 | 69.3±0.9 |
| | Adapt-patch | 78.2±1.2 | 81.4±1.4 | 75.1±0.8 | 82.4±2.7 | 86.7±1.8 | 83.1±1.1 |
| SS | BadNet | 0.6±0.2 | 3.7±1.3 | **6.5±0.8** | 0.7±0.2 | 4.5±1.8 | **6.9±0.9** |
| | Blend | 0.7±0.3 | 2.6±1.5 | **5.2±1.1** | 1.6±0.7 | 3.5±1.1 | **5.7±0.5** |
| | Adapt-blend | **7.3±1.7** | 4.8±1.3 | 5.7±0.7 | 12.8±1.9 | 11.7±1.3 | **15.6±0.7** |
| | Adapt-patch | 9.5±2.1 | 10.9±1.7 | **14.2±1.2** | 10.5±2.1 | 11.3±1.2 | **14.9±0.3** |
| AC | BadNet | 0.4±0.1 | 7.5±1.2 | **10.1±0.6** | 2.6±0.9 | 8.2±1.6 | **11.4±1.1** |
| | Blend | 0.2±0.1 | 9.3±2.3 | **11.9±1.7** | 3.4±1.5 | 7.6±1.2 | **9.7±0.8** |
| | Adapt-blend | 10.2±2.5 | 18.7±2.1 | **23.5±1.6** | 3.4±2.3 | 4.2±1.8 | **6.7±0.7** |
| | Adapt-patch | 13.5±2.1 | 21.7±1.3 | **26.8±1.0** | 5.2±1.6 | 5.7±1.2 | **7.4±0.9** |
| SCAn | BadNet | **85.5±3.8** | 84.9±3.2 | 83.2±2.1 | 78.3±2.9 | 77.6±2.1 | **81.9±1.4** |
| | Blend | **84.1±1.6** | 83.5±1.2 | 82.9±0.8 | 80.2±2.1 | **81.4±1.3** | 80.9±0.9 |
| | Adapt-blend | 69.7±2.7 | 68.7±1.8 | **72.6±1.1** | 68.8±3.4 | **69.4±1.6** | 67.9±1.5 |
| | Adapt-patch | 71.7±1.5 | 71.3±0.9 | **73.9±0.7** | 81.9±2.7 | 81.2±1.6 | **82.1±1.1** |
| STRIP | BadNet | 72.3±2.7 | 71.8±1.8 | **77.1±1.2** | 67.6±3.2 | 68.1±2.4 | **73.7±1.3** |
| | Blend | **83.2±3.2** | 82.9±2.5 | 82.8±1.6 | 71.9±2.7 | 71.2±1.6 | **75.1±0.9** |
| | Adapt-blend | 64.4±3.7 | 67.9±2.3 | **70.6±1.6** | 69.2±2.8 | **70.8±1.5** | 68.5±0.7 |
| | Adapt-patch | 67.8±2.5 | 67.5±1.7 | **72.7±1.3** | 74.7±1.9 | **75.4±1.3** | 73.5±0.8 |
| SPECTRE | BadNet | 0.2±0.1 | 3.9±1.4 | **7.3±0.6** | 0.6±0.2 | 2.5±0.7 | **4.1±0.5** |
| | Blend | 0.6±0.2 | 12.4±1.5 | **14.7±0.5** | 9.5±1.4 | 12.5±1.3 | **14.7±0.9** |
| | Adapt-blend | 14.8±1.5 | 19.6±1.3 | **20.3±0.9** | 15.7±2.3 | 16.9±1.7 | **20.1±1.2** |
| | Adapt-patch | 17.9±2.1 | 25.8±1.4 | **27.3±0.8** | 19.3±1.9 | 20.5±1.3 | **21.6±0.7** |
| ABL | BadNet | 9.3±2.4 | 13.9±1.7 | **17.4±0.7** | 5.7±1.3 | 9.6±1.5 | **10.2±1.1** |
| | Blend | 20.8±2.7 | 22.7±1.3 | **25.7±1.1** | 59.1±2.7 | 58.3±2.1 | **62.6±1.4** |
| | Adapt-blend | 23.7±2.5 | 23.2±1.5 | **25.8±0.8** | 43.3±3.2 | 44.8±2.7 | **46.4±1.6** |
| | Adapt-patch | 19.8±1.8 | 20.4±1.2 | **21.9±1.0** | 45.8±2.8 | 45.2±1.7 | **47.9±1.3** |
| FP | BadNet | 29.4±2.7 | 30.1±1.4 | **35.3±0.9** | 61.8±3.5 | 63.7±2.1 | **64.1±1.6** |
| | Blend | 67.2±2.8 | 68.1±2.3 | **71.1±1.1** | 73.1±2.9 | 72.7±1.8 | **74.2±1.3** |
| | Adapt-blend | 60.7±1.5 | 57.3±1.1 | **62.6±0.8** | 69.7±3.1 | 70.3±2.5 | **73.4±1.4** |
| | Adapt-patch | 66.3±2.4 | 64.1±1.9 | **69.7±1.2** | 70.1±2.5 | **69.7±1.8** | 69.5±1.5 |
| NC | BadNet | 35.6±3.4 | 42.1±2.9 | **52.4±1.4** | 43.7±3.2 | 44.8±2.5 | **49.5±0.8** |
| | Blend | 78.1±2.5 | **79.4±1.8** | 77.2±1.3 | 68.4±2.4 | 69.2±1.6 | **72.3±1.1** |
| | Adapt-blend | 66.9±1.7 | 64.2±1.3 | **70.3±0.9** | 66.2±2.7 | 65.4±1.4 | **67.8±0.6** |
| | Adapt-patch | 18.3±1.3 | 19.5±0.9 | **23.6±0.4** | 2.7±0.7 | 4.1±1.2 | **4.6±0.8** |

## 8.9 Ablation study on poisoning rate

We conduct additional experiments on Cifar10 dataset, ResNet18 model and backbone attack BadNet to test how different poisoning rate affect the proposed method. We report the success rate against undefended models and three representative defenses in Table 13. According to the results, poisoning more samples can increase the success rate against undefended models, and slightly increase the poisoning effect against defenses. We notice that while the poisoning rate is increasing, the improvement of the poisoning effect against defenses becomes minor. This can be because when the number of poisoned samples increases, samples that are farther from the boundary are included. These samples can achieve better performance when there is no defense but are easy to detect. This also highlights the importance of a proper sample selection strategy in backdoor attacks.

## 8.10 Discussion on surrogate models

Our experiments in Table 1 shows that the poisoned samples generated from ResNet18 can be transferred well to VGG16. We further check whether different surrogate models select very different samples. We

Table 8: Full Performance on Type II backdoor attacks.

| | Model Defense | Attacks | ResNet18 | | | ResNet18 → VGG16 | | |
|---|---|---|---|---|---|---|---|---|
| | | | Random | FUS | CBS | Random | FUS | CBS |
| CIFAR10 | No Defenses | Hidden-trigger | 81.9±1.5 | 84.2±1.2 | 76.3±0.8 | 83.4±2.1 | 86.2±1.3 | 79.6±0.7 |
| | | LC | 90.3±1.2 | 92.1±0.8 | 87.2±0.5 | 91.7±1.4 | 93.7±0.9 | 87.1±0.8 |
| | NC | Hidden-trigger | 6.3±1.4 | 5.9±1.1 | **8.7±0.9** | 10.7±2.4 | 11.2±1.5 | **14.7±0.6** |
| | | LC | 8.9±2.1 | 8.1±1.6 | 12.6±1.1 | 11.3±2.6 | 9.8±1.1 | 12.9±0.9 |
| | SS | Hidden-trigger | 68.5±3.2 | 69.3±2.4 | **74.1±1.3** | 75.7±3.1 | 74.8±2.3 | **76.2±1.1** |
| | | LC | **87.2±1.3** | 86.6±0.8 | 86.9±0.5 | 85.4±2.7 | **85.5±1.8** | 84.2±1.2 |
| | FP | Hidden-trigger | 11.7±2.6 | 9.9±1.3 | **14.3±0.9** | 8.6±2.4 | 8.1±1.4 | **11.8±0.8** |
| | | LC | 10.3±2.1 | 13.5±1.2 | **20.4±0.7** | 7.9±1.7 | 8.2±1.1 | 10.6±0.7 |
| | ABL | Hidden-trigger | 1.7±0.8 | 5.6±1.6 | **10.5±1.1** | 3.6±1.1 | 8.8±0.8 | **10.4±0.6** |
| | | LC | 0.8±0.3 | 8.9±1.5 | **12.1±0.8** | 1.5±0.7 | 9.3±1.2 | **12.6±0.8** |
| CIFAR100 | No Defenses | Hidden-trigger | 80.6±2.1 | 84.1±1.8 | 78.9±1.3 | 78.2±2.3 | 81.4±1.6 | 75.8±1.2 |
| | | LC | 86.3±2.3 | 87.2±1.4 | 84.7±0.9 | 84.7±2.8 | 85.2±1.4 | 81.5±1.1 |
| | NC | Hidden-trigger | 3.8±1.4 | 4.2±0.9 | **7.6±0.7** | 4.4±1.1 | 5.1±1.2 | **6.8±0.9** |
| | | LC | 6.1±1.8 | 5.4±1.1 | **8.3±0.5** | 3.9±1.2 | 3.8±0.9 | **8.3±0.7** |
| | SS | Hidden-trigger | 72.5±2.6 | 71.9±1.7 | **74.7±1.2** | **75.3±3.1** | 74.8±2.1 | 73.1±1.3 |
| | | LC | **80.4±2.4** | 80.1±1.4 | 79.6±1.3 | 82.9±2.7 | **83.5±1.8** | 81.4±1.0 |
| | FP | Hidden-trigger | 15.3±3.1 | 16.7±0.9 | **18.2±0.7** | 8.9±1.3 | 9.3±1.1 | **10.3±0.7** |
| | | LC | 13.8±2.7 | 12.7±1.5 | **14.9±0.6** | 10.3±1.4 | 9.9±0.8 | **12.2±0.5** |
| | ABL | Hidden-trigger | 2.3±0.9 | 3.9±1.3 | **6.5±1.1** | 3.7±0.9 | 3.5±0.7 | **6.4±0.4** |
| | | LC | 0.9±0.2 | 2.7±0.8 | **6.2±1.2** | 2.5±0.8 | 2.1±0.7 | **6.7±0.5** |

Table 9: Accuracy on clean input of each backdoored model with all three sample selections.

| | Attack | ResNet18 | | | VGG16 | | |
|---|---|---|---|---|---|---|---|
| | | Random | FUS | CBS | Random | FUS | CBS |
| Cifar10 | BadNet | 92.5 | 93.2 | **95.1** | 90.1 | 90.4 | **91.3** |
| | Blend | 93.7 | 93.6 | **94.8** | 91.6 | 91.8 | **92.5** |
| | Adapt-blend | 93.2 | 93.7 | **94.5** | 91.8 | 91.4 | **92.2** |
| | Adapt-patch | 92.7 | 92.9 | **93.8** | 91.4 | 91.5 | **92.3** |
| | Hidden-trigger | 94.6 | 94.3 | **95.7** | 92.7 | 93.0 | **93.6** |
| | LC | 93.5 | 94.2 | **94.9** | 92.8 | 92.5 | **93.4** |
| | Lira | 94.2 | 94.1 | **95.1** | 92.6 | 92.8 | **93.3** |
| | WaNet | 94.3 | 94.7 | **95.5** | 92.7 | 93.1 | **93.4** |
| | WB | 93.9 | 94.2 | **94.9** | 92.6 | 92.7 | **93.2** |
| Cifar100 | BadNet | 75.2 | 76.1 | **76.8** | 72.2 | 72.8 | **73.4** |
| | Blend | 76.9 | 77.3 | **77.7** | 73.4 | 73.6 | **74.2** |
| | Adapt-blend | 76.4 | 76.1 | **76.9** | 72.8 | 73.1 | **73.8** |
| | Adapt-patch | 77.1 | 76.8 | **77.9** | 72.9 | 72.7 | **73.4** |
| | Hidden-trigger | 77.6 | 77.6 | **78.2** | 73.7 | 73.8 | **74.2** |
| | LC | 76.8 | 77.1 | **77.6** | 73.6 | 73.7 | **73.9** |
| | Lira | 77.6 | 77.4 | **78.1** | 73.8 | 73.6 | **74.3** |
| | WaNet | 77.7 | 78 | **78.3** | 73.8 | 73.5 | **74.1** |
| | WB | 77.5 | 77.3 | **78.2** | 73.5 | 73.6 | **74.3** |

compare ResNet18 and VGG16, and these two models have similar clean accuracy on the Cifar10 dataset (95.5% and 93.6% respectively). We fix class 1 as the target class and select 100 samples using CBS from both models. We notice that 67% of selected samples are the same, which is quite a large proportion. This can explain why our method can transfer well from ResNet18 to VGG16.

Table 10: Performance of three types of backdoor attacks on Tiny-ImageNet dataset.

| | | Attacks | Random | FUS | CBS |
|---|---|---|---|---|---|
| **Type I** | No defenses | **BadNet** | 89.5+0.8 | **89.8+0.2** | 83.1+0.6 |
| | | **Blended** | 83.4+1.2 | **85.2+0.3** | 81.6+0.5 |
| | | **Adaptive-Blend** | 67.2+0.7 | **68.9+0.4** | 66.2+0.7 |
| | | **Adaptive-Patch** | 84.5+1.1 | **86.3+0.3** | 81.7+0.5 |
| | SS | **BadNet** | 0.4+0.2 | 10.8+0.1 | **18.5+0.1** |
| | | **Blended** | 37.2+1.2 | 43.2+0.6 | **46.3+0.8** |
| | | **Adaptive-Blend** | 59.4+0.9 | 61.7+0.4 | **65.1+0.6** |
| | | **Adaptive-Patch** | 75.3+1.3 | 76.5+0.2 | **78.5+0.4** |
| | Strip | **BadNet** | 0.6+0.3 | 5.1+0.1 | **12.2+0.2** |
| | | **Blended** | 46.2+1.7 | 50.9+0.3 | **54.6+0.6** |
| | | **Adaptive-Blend** | 58.4+1.2 | 61.4+0.5 | **63.3+0.4** |
| | | **Adaptive-Patch** | 69.5+1.1 | 71.2+0.4 | **72.8+0.5** |
| **Type II** | No defenses | **Hidden-trigger** | 59.7+0.8 | **62.9+0.3** | 54.3+0.3 |
| | NC | **Hidden-trigger** | 5.3+0.7 | 8.5+0.3 | **11.5+0.2** |
| | FP | **Hidden-trigger** | 8.4+0.9 | 9.7+0.2 | **12.1+0.4** |
| | ABL | **Hidden-trigger** | 1.8+0.6 | 2.6+0.2 | **4.2+0.4** |
| **Type III** | No defenses | **WaNet** | 98.5+0.5 | **99.2+0.2** | 96.1+0.3 |
| | | **LiRA** | 99.3+0.6 | **99.7+0.1** | 96.4+0.4 |
| | | **WB** | 98.2+0.4 | **99.5+0.2** | 92.7+0.2 |
| | NC | **WaNet** | 5.4+0.8 | **11.2+0.4** | 10.7+0.3 |
| | | **LiRA** | 6.3+1.1 | 9.7+0.3 | **10.2+0.5** |
| | | **WB** | 9.6+0.8 | 13.5+0.5 | **15.1+0.4** |
| | FP | **WaNet** | 9.5+0.9 | 12.6+0.2 | **13.6+0.4** |
| | | **LiRA** | 8.7+1.2 | 10.7+0.3 | **13.8+0.3** |
| | | **WB** | 10.2+0.8 | 13.1+0.4 | **16.5+0.6** |

Table 11: Additional results on ImageNet-1k and GTSRB dataset.

| Attack | Defense | ImageNet-1k | | GTSRB | |
|---|---|---|---|---|---|
| | | Random | CBS | Random | CBS |
| **BadNet** | No defense | **92.5** | 90.2 | **90.6** | 89.4 |
| | NC | 9.6 | **21.7** | 1.8 | **22.3** |
| | SS | 6.9 | **19.3** | 1.2 | **18.4** |
| **Hidden trigger** | No defense | **83.2** | 80.5 | **78.5** | 75.3 |
| | NC | 21.3 | **30.5** | 10.3 | **15.7** |
| | FP | 32.6 | **39.1** | 19.8 | **24.7** |
| **WaNet** | No defense | **98.1** | 94.3 | **98.6** | 96.4 |
| | NC | 15.9 | **25.2** | 17.4 | **23.8** |
| | SS | 80.5 | **84.2** | 83.1 | **89.3** |

## 8.11 Discussion on the difference between raw input space and latent space

CBS is based on the observation that poisoned samples can be separate from the target class in the latent space, which raises a question of whether detecting outliers in the raw space can defend against such an attack. To investigate, we compare the distance between each sample and the center of its class in both

Table 12: Two strategies to improve the poisoning performance when no defenses. CBS+1 and CBS+2 denote two strategies respectively.

|  | Attack | Random | FUS | CBS | CBS+1 | CBS+2 |
|---|---|---|---|---|---|---|
| No defense | BadNet | 99.9 | 99.9 | 93.6 | 96.3 | 98.5 |
|  | Blend | 89.7 | 93.1 | 86.5 | 87.3 | 88.9 |
| SS | BadNet | 0.5 | 4.7 | 20.2 | 18.8 | 17.5 |
|  | Blend | 43.7 | 42.6 | 55.7 | 53.2 | 51.8 |
| STRIP | BadNet | 0.5 | 8.5 | 23.7 | 22.3 | 20.9 |
|  | Blend | 54.7 | 57.2 | 60.6 | 59.1 | 58.4 |
| ABL | BadNet | 16.8 | 17.3 | 31.3 | 29.6 | 27.7 |
|  | Blend | 57.2 | 55.1 | 65.7 | 63.5 | 61.8 |
| SPECTRE | BadNet | 0.9 | 10.1 | 19.6 | 17.5 | 16.4 |
|  | Blend | 9.2 | 16.7 | 24.2 | 22.1 | 21.3 |

Table 13: Ablation study on poisoning rates. Test on ResNet18 model, Cifar10 dataset and BadNet attack.

| Poison rate | 0.1%(50) | 0.2%(100) | 0.5%(250) | 1%(500) |
|---|---|---|---|---|
| No defense | 91.8 | 93.6 | 96.7 | 98.4 |
| SS | 18.7 | 20.2 | 23.5 | 24.2 |
| ABL | 27.4 | 31.3 | 33.1 | 34.8 |
| NC | 23.7 | 24.6 | 28.4 | 30.6 |

raw input space and latent space, i.e. $d_{raw}(x) = \|x - x_{y,mean}\|_2$ and $d_{latent}(x) = \|f(x) - f(x)_{y,mean}\|_2$ where $f(x)$ denote the latent representation w.r.t the model $f$. We use the backdoored ResNet18 model and Cifar10 dataset. Then we plot two distances in the same figure to check their relationship, shown in Figure 11. Based on the result, there is no obvious relationship between them, and the outliers in the raw space can have a very small distance to the center in the latent space. This indicates that when filtering out the outliers based on distances in the raw space, the poisoned samples can still be preserved.

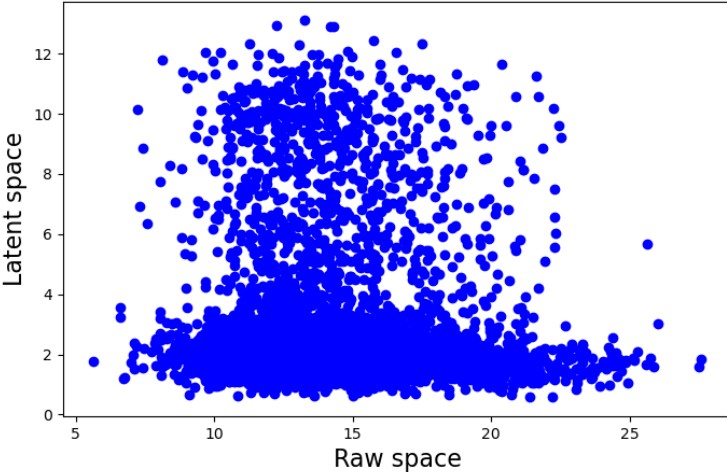

Figure 11: Relationship between the distance in the raw input space and latent space.

To further validate, we consider a naive defense: ignore data points $(x, y)$ from the training set if $\|x - x_{y,mean}\|_2 \geq \alpha$ where $x_{y,mean} := mean\{x'|(x', y) \in data\}$. We follow Table 1 and test on Cifar10 and ResNet18. We remove samples with a l2 distance in the raw space larger than 18, which is about 12% of total samples. Then we train the model on the filtered dataset and test the success rate. We also consider BadNet and Blended as the backbone for illustration.

Table 14: Test on naive defenses that remove outliers in the raw input space.

|  | Attack | CBS |
|---|---|---|
| **No defense** | **BadNet** | 93.6 |
|  | **Blend** | 86.5 |
| **Naive defense** | **BadNet** | 90.7 |
|  | **Blend** | 84.2 |

As shown in Table 14, the success rate after the naive defense does not drop much, which suggests that this defense can not effectively defend the proposed attack. Due to the difference between the raw input space and the latent space, the outliers in the raw input space may not be the actual poisoned samples.

