# OpenReview forum: "Stealthy Backdoor Attack via Confidence-driven Sampling"
_TMLR — Accepted by TMLR_

### Review · Reviewer_X7j2 · 2024-09-01

**Summary Of Contributions:**

The paper introduces Confidence-driven Boundary Sampling (CBS), a novel method that enhances the stealthiness of backdoor attacks in deep neural networks by strategically selecting samples near the decision boundary, identified through low confidence scores. This approach addresses the limitations of random sampling, which often makes attacks more detectable. CBS is versatile, theoretically supported, and can be integrated with various existing backdoor attack strategies, improving their stealthiness with minimal impact on attack success rates. Extensive experiments validate the effectiveness of the proposed method under different defense mechanisms.

**Audience:**

No

**Broader Impact Concerns:**

Not involved.

**Claims And Evidence:**

No

**Requested Changes:**

Please see weakness.

**Strengths And Weaknesses:**

Strengths
1.	The paper provides a solid theoretical analysis to support the proposed method, enhancing the credibility of the approach.
2.	The paper is well-structured, with clear explanations of the problem, proposed solution, and experimental results.
Weaknesses
1.	The relationship between the poison rate and the effectiveness of CBS is not deeply explored, leaving some uncertainty about how the method scales with varying levels of poisoning.
2.	While CBS is presented as efficient, the paper could benefit from a more detailed discussion on the computational overhead compared to random sampling, especially for large-scale datasets.
3.	The experiments primarily focus on standard datasets like CIFAR-10, which might not fully represent the method's performance in more complex or real-world scenarios. Testing on a wider variety of datasets would strengthen the evaluation.

---

> ### Author Response · Authors · 2024-09-25
> **Response to Reviewer X7j2**
>
> We thank the reviewer for the valuable comments, and we would like to address the concerns respectively.
>
> **1. Discussion of poisoning rates.**
>
> We thank the reviewer for raising this comment. We conduct additional experiments on the Cifar10 dataset, ResNet18 model, and backbone attack BadNet to test how different poisoning rates affect the proposed method. We report the success rate against undefended models and three representative defenses.
>
> | Poison rate | 0.1%(50) | 0.2%(100) | 0.5%(250) | 1%(500) |
> |-------------|----------|-----------|-----------|---------|
> | No defense  | 91.8     | 93.6      | 96.7      | 98.4    |
> | SS          | 18.7     | 20.2      | 23.5      | 28.2    |
> | ABL         | 27.4     | 31.3      | 33.1      | 39.8    |
> | NC          | 23.7     | 24.6      | 28.4      | 35.6    |
>
> According to the results, poisoning more samples can increase the success rate against undefended models, and slightly increase the poisoning effect against defenses. We notice that while the poisoning rate is increasing, the improvement of the poisoning effect against defenses becomes minor. This can be because when the number of poisoned samples increases, samples that are farther from the boundary are included. These samples can achieve better performance when there is no defense but are easy to detect. This also highlights the importance of a proper sample selection strategy in backdoor attacks. We include this discussion in the revised version, **Page 30, section 7.9**, and mentioned in the main text in **Page 14**.
>
> **2. Discussion of computation overhead**
>
> We thank the reviewer for raising this concern. We analyze the computation overhead as follows. Suppose the sample size is $N$, poison rate is $r$. For the random selection, we need to randomly select $rN$ samples from the dataset whose time complexity is $O(rN)$. For the proposed method, we need to first compute the confidence score for each sample and then select those smaller than $\epsilon$. Suppose the average time for computing the confidence score for one sample is $t$, and the time complexity for the proposed method is $O(N(t+1))$. Therefore, both complexity is linear in order of sample size $N$. We include this analysis on **Page 25, Section 7.4** of the revision and mentioned in the main text **Page 5**.
>
> **3. Additional experiments on other datasets.**
>
> We thank the reviewer for raising this concern. We consider two additional datasets, ImageNet-1k which is a large-scale image dataset consisting of 1000 object classes and more than 1,200,000 images, and GTSRB which is a traffic sign recognition benchmark. To illustrate the effectiveness of the proposed method, we test representative backbone attacks from all three types and representative defenses. Since FUS is time-consuming and not eligible for large datasets, we only compare it with the random baseline.
>
> | Attack         | Defense    | ImageNet-1k |      | GTSRB  |      |
> |----------------|------------|-------------|------|--------|------|
> |                |            | Random      | CBS  | Random | CBS  |
> | BadNet         | No defense | 92.5        | 90.2 | 90.6   | 89.4 |
> |                | NC         | 9.6         | 21.7 | 1.8    | 22.3 |
> |                | SS         | 6.9         | 19.3 | 1.2    | 18.4 |
> | Hidden trigger | No defense | 83.2        | 80.5 | 78.5   | 75.3 |
> |                | NC         | 21.3        | 30.5 | 10.3   | 15.7 |
> |                | FP         | 32.6        | 39.1 | 19.8   | 24.7 |
> | WaNet          | No defense | 98.1        | 94.3 | 98.6   | 96.4 |
> |                | NC         | 15.9        | 25.2 | 17.4   | 23.8 |
> |                | SS         | 80.5        | 84.2 | 83.1   | 89.3 |
>
> According to these results, our method is effective on various real-world tasks and large-scale datasets. We include this discussion in the revised paper, **Page 27, section 7.7** and mentioned in the main text **Page 10**.

---

> ### Author Response · Authors · 2024-09-27
> **A friendly reminder**
>
> We are grateful for your comments. We carefully revise the paper and provide detailed responses to address all your concerns. Please let us know if you have further concerns. We are looking forward to more discussion.

---

### Review · Reviewer_RvbD · 2024-09-03

**Summary Of Contributions:**

This article investigates the role of sampling in the effectiveness of backdoor attack strategies. The authors advocate a sampling algorithm that prioritizes inputs if a surrogate model learned on clean data is uncertain on the predicted label. As a motivation, the authors suggest that their approach could help break defense strategies that rely on outlier detection in the latent representation space. Experimental evidence shows that CBS (confidence-driven boundary sampling) leads to stronger backdoor attacks on CIFAR10, CIFAR100, and TinyImagenet.

**Audience:**

Yes

**Broader Impact Concerns:**

Since the nature of this work concerns the safety of AI systems. I suggest the authors add a small section on the broader impact of designing efficient backdoor attacks.

**Claims And Evidence:**

No

**Requested Changes:**

**Potential Improvements**
- Figure 1 suggests that CBS enables a mixing of the clean and backdoored samples in the latent representation space. Is this also true for the raw input space? Consider a backdoor defense algorithm that ignores data points (x,y) from the training dataset if $\|\|x - x_{y, median}\|\|\_2 \geq \alpha$
where $x_{y, median} := \mathrm{median} \left(x’  | (x’, y) \in \text{ backdoored training data } \right)$. Intuitively this defense algorithm aims to detect outliers in raw-pixel space rather than learnt representations. Can the authors comment on how CBS will work against this naive strategy?
- One intuitive argument that could favor CBS vs random sampling is the clean accuracy of backdoored models on clean data. Intuitively one might expect that the clean accuracy is higher for CBS-based backdoored models since only data closer to the decision boundaries are modified. Can the authors report the clean accuracy of backdoored models across the 3 sampling choices - Random, FUS and CBS?


**Technical Clarifications**

- _Attack stealthiness_ is not formally defined. Even if the idea appears intuitive there is value in presenting a formal definition of this phenomena as either a separation in latent space or identifiability by a specific class of outlier-detection based defense algorithms. I understand that this is an empirical phenomena but a lack of formal definition obfuscates the inferences from resulting experiments.
- “_As seen in Figures 2a and 2b, random sampling is likely to select samples that are close to the center of their true classes_” Why do the figures have negative points on the x-axis? Isn’t the x-axis a distance measurement?
- The authors should clarify what the image generator $G_t (\cdot)$ guarantees concerning trigger injection. For eg, how does the trigger $t$ manifest in the generated image? I assume $G_t(x)$ cannot be an identity function as that would ignore the trigger $t$, are there any formal constraints? This would help formalize the difference between a clean dataset and a trigger-embedded dataset.
- “_To elaborate, we further calculate the distance (1) from each selected sample (without trigger) to the center (2) of their true classes computed on the clean model, which is denoted as $d_o$_” I request the authors to please explicitly define the distance $d_o$ rather than the informal description in the footnotes.
- “_Formally, we define the distance between each selected sample (with trigger) and the center of the target class computed on the backdoored model, as $d_t$_”. Here the authors refer to the representation of the selected samples rather than samples themselves and similarly the authors refer to the average of the representations of samples with class - target label rather than the target class. It is unclear why the reader has to infer these instead of being presented with explicit definitions. At first read, there is a clear type mismatch in measuring distance between samples and a class label.
- Page 16, The authors write $\mathbb{E}\_{x \sim C_i}$ by which they presumably mean $\mathbb{E}\_{x \sim \mathrm{Uniform}(C\_i) }$ i.e, the expectation is taken w.r.t the discrete uniform distribution on the finite set of vectors $C_i$.
- Page 17, “select $x_b$ such that … $= \min_{x \in C\_1} \|\|x- \hat{\mu}_2\|\|_2$, What is $\hat{\mu}_2$ do the authors mean $\tilde{\mu}_2$?

**Minor Editorial Clarifications**

- The following sentence is unclear and also has a typo “_Except for the promising results due to trigger designs in existing literature, sampling methods that select **poisonined** data to add triggers are attracting more attention._”
- The following sentence is unclear due to _type mismatch_ (as explained above), “_Second, the closer a sample is from its true class on the clean model, the farther it gets from the target class on the backdoored model._” The sentence appears to compare the distance of a sample from a class, presumably the authors mean the distance of the same from the region assigned label c, if so, then there needs to be clearer formal definitions.
- The following two sentences appear to convey the same idea. If I’m misunderstanding can the authors clarify the difference? “_For any given clean test set $D\_{te}$, the accuracy of $f(·; θ\_b)$ evaluated on trigger-embedded dataset $T(D\_{te})$ is referred to as success rate, and attackers also expect to see high success rate on any clean samples with triggers embedded._”
- In the definition of Trigger injection in Section (3.2), “$G\_t(x)$ _is the attacker-specified poisoned image generator with trigger pattern $t$”. I think it will be helpful if the authors differentiate between the image generator function $G\_t$  and the generated image $G\_t(x)$. This will be consistent with the description of $S$ (instead of $S(x,y)$) as the target label generator.
- Typo : “_This indicates a separation in latent space which can be easily detected by potential defenses. For example, Spectral **Siginiture** …_”
In page 16 and 17, I think the authors mean $d^2\_M(\tilde{x} , \tilde{C}\_2)$ rather than $d^2\_M(\tilde{x} , \tilde{D}\_2)$  or $d^2\_M(\tilde{x} , \tilde{C}\_{2,x})$.

**Strengths And Weaknesses:**

**Strengths**
The core idea is intuitive. Poisoning samples closer to the decision boundary inflates the hardness of the learning task.


This reader perceives two important issues that the authors need to address.

**Weakness 1 : Threat model.**
The threat model is weaker than one might think at first sight. The threat model requires access to the full training dataset. In Section 3.1, the following two consecutive sentences that discuss the threat model appear to be different scenarios.
1)  “We assume that the attacker has access to the clean training set and can modify a proportion of the training data. Then the victim trains his own models on this data and the attacker has no knowledge of this training procedure.”
2) “In a real-world situation, attackers can upload their datasets to the Internet. They can sneakily insert backdoors into their data and then share it with victims, who unknowingly use it to train their own models (Gu et al., 2017; Chen et al., 2017).”

The authors should be clearer about stating that their threat model defined in (1) above is not the same as the threat model discussed in (2).
- The threat model requires access to a surrogate model trained on clean samples.
Currently, the impact of surrogate models is restricted to evaluation of the overall success rate of the adversarial attack. Can the authors discuss the effect of the surrogate model on the choice of poisoned samples. I.e., What is the variability in the choice of poisoned samples across different choices of surrogate models?
- In Theorem 4.2, the authors assume that the attacker is aware of the mean of the clean data distribution (so that one can ensure that the trigger satisfies the constraint $(\mu_2-\mu_1)^Tt \geq 0$  This is stronger than assuming access to i.i.d samples from the clean data distribution. Please explicitly mention that this is stronger than the threat model considered.

**Weakness 2 : Informal presentation.**
The lack of formal definitions makes the writing unclear.

- There is likely a version Theorem 4.2 that is correct and precise and makes a similar argument in spirit. However, the current statement and its proof arguments lack precision and ignore necessary details on the effect of randomness of sampling $C_1, C_2$ and $\tilde{x}$.
- Theorem 4.2 assumes a hard-margin exists. As a reader I would like to know when the hard-margin exists for a fixed trigger t that satisfies the constraint w.r.t $\mu_1$ and $\mu_2$? Presumably, this is a probability event over the randomness of the samples $C_1, C_2$ and the choice of input being targeted. Can the authors clarify and quantify this? In particular, the choice of trigger might restrict the choice of input that can be modified? It is unclear whether this allows for a uniformly random choice or CBS choice in the context of the SVM example.
- Page 17, “Let $n\rightarrow \infty$, we can obtain the result.”  The set $C_1$ and $C_2$ are random samples. Do the authors mean $$\lim_{n \rightarrow \infty} \mathbb{E}_{C_1 \sim p_1, C_2 \sim p_2, |C_1| = |C_2| = n} \ldots?$$ To complete this argument one needs a prescription of the choice of $x \in C_1$ that is modified with trigger t. Further, the choice of $x$ is random since $C_1$ is random and hence $\|\|\tilde{x}\|\|_2$ is also random.
- The authors should explicitly state the following distances (while keeping track of randomness of $C_1, C_2, \tilde{x}$), $$d_M(\tilde{x}_b, \tilde{C}_2), \text{ and } d_M(\tilde{x}_u, \tilde{C}_2)$$
- Re: Remark 4.4, why include the $n \rightarrow \infty$ in the theorem statement if it does not change the inference (given the above additional technical considerations)?

---

> ### Author Response · Authors · 2024-09-25
> **Response to Reviewer RvbD 1/**
>
> We thank the reviewer for the valuable comments. We would like to address all concerns respectively.
>
> **1. Discussion of the threat model**
>
> We thank the reviewer for raising this concern.
> * For the two sentences
> 1. "We assume that the attacker has access to the clean training set..."
> 2. "In a real-world situation, attackers can upload their datasets..."
>
> We modify the second statement to make it consistent with the first one. In a real-world situation, the attacker can access some clean datasets, and modify a proportion of them by inserting triggers to them. Then they upload the poisoned samples to the Internet and victims unknowingly download and use it for training. Therefore, the attacker does not know the model or training algorithm used by the victims. We include this modification in revision, **Page 3 Section 3.1.**
>
> * Surrogate model: To provide more concrete evidence on why surrogate models work, we take two surrogate models ResNet18 and VGG16 as examples for discussion. These two models have similar clean accuracy on the Cifar10 dataset (95.5% and 93.6% respectively). We fix class 1 as the target class and select 100 samples using CBS from both models. We notice that 67% of selected samples are the same, which is quite a large proportion. This can explain why our method can transfer well from ResNet18 to VGG16. We would like to also clarify that we leverage the surrogate model because we assume the attacker does not have information about the exact model used by the victim. We add this discussion into the revised paper, **Page 30 Section 7.10** and mention it in the main content **Page 10**.
>
> * Trigger assumption in Theorem 4.2: This assumption is to guarantee that the trigger moves the poisoned sample towards the target class, which indeed aligns with the common practice in real-world backdoor attacks [1]. We also note that, with the significant changes in Section 4.3, we provide detailed discussions on how the trigger $t$, the sample $x$, and the hard margin of SVM are related in **Remark 4.4** in the revision.
>
> **2. Revision of Theorem 4.2**
>
> We appreciate the reviewer providing constructive comments to Theorem 4.2. To address the concerns, we revise the second half of Section 4.3 to provide more detailed results:
> * Theorem 4.2: A finite-sample analysis of the Mahalanobis distance from any fixed $\tilde{x}$ to the cluster of samples in $C_2$. We provide the probability bound and show that the distance converges. In addition, the support vector in $C_1$ has the minimal (population) distance to $C_2$, and based on Proposition 4.6 later, CBS is likely to select a sample that is close to this support vector.
> * Theorem 4.3: The analysis of the attack success rate when there are infinite samples for any fixed $\tilde{x}$. This part follows the analysis in the original Theorem 4.2.
> * Remark 4.4: To clarify how the trigger and the selected sample affect the margin of SVM, we provide Remark 4.4 (for the population version) and later supplement with Proposition 4.6 and 4.7 for the finite-sample scenario. For CBS, as long as $t^T(-\mu_1)>0$, a hard margin exists. For random selection, there are more restrictions on the choice of $t$ depending on the exact choice of $x$.
> * Theorem 4.5: A finite-sample analysis of the margin. Since there are infinitely many choices of the hyperplane which can separate $C_1$ and $C_2$, we study the margin of these hyperplanes to finite samples compared to their margin to the population (before injecting the poisoned sample). We provide a probability bound and show that the finite-sample margin converges to the corresponding limit uniformly for all hyperplanes.
> * Proposition 4.6 and 4.7: There are two implications: (1) the sample CBS selected is close to the support vector in the population, and (2) after perturbing the sample selected by CBS/random selection, the new margins are also close to those obtained in Theorem 4.3.
>
> **References**
>
> [1] BadNets: Identifying Vulnerabilities in the Machine Learning Model Supply Chain. T Gu et al, 2017.

---

> ### Author Response · Authors · 2024-09-25
> **Response to Reviewer RvbD 2/**
>
> **3. A discussion of the proposed defense**
>
> We thank the reviewer for proposing this defense. Following your suggestion, we compare the distance between each sample and the center of its class in both raw input space and latent space, i.e. $d_{raw}(x)=\|x-x_{y,mean}\|_2 $
>
> and $d_{latent}(x)=\|f(x)-f(x)_{y,mean}\|_2$ where $f(x)$ denote the latent representation w.r.t the model $f$.
> We use the backdoored ResNet18 model and Cifar10 dataset. Then we plot two distances in the same figure to check their relationship (shown in the revised paper, Figure 11, because we can not insert images here). Based on the result, there is no obvious relationship between the distance in the raw space and the latent space, and the outliers in the raw space can have a very small distance to the center in the latent space. This indicates that when filtering out the outliers based on distances in the raw space, the poisoned samples may not be efficiently removed. We attach this plot and include the discussion in the revision **Page 32, Section 7.11** and mention it in the main content **Page 4**.
>
> We further conduct experiments applying this defense. We follow Table 1 in the paper and test on Cifar10 and ResNet18. We remove samples with a l2 distance in the raw space larger than 18, which is about 12% of the total samples. Then we train the model on the filtered dataset and test the success rate. We also consider BadNet and Blend as the backbone for illustration.
>
> |               |        | CBS  |
> |---------------|--------|------|
> | No defense    | BadNet | 93.6 |
> |               | Blend  | 86.5 |
> | Naive defense | BadNet | 90.7 |
> |               | Blend  | 83.2 |
>
> As shown in this table, the success rate after the naive defense does not drop much, which suggests that this defense can not effectively defend the proposed attack. Due to the difference between the raw input space and the latent space, the outliers in the raw input space may not be the actual poisoned samples.
>
> **4. Clean accuracy**
>
> We have included the clean accuracy in the revised paper, **Page 27, Section 7.7** and mention it in the main content on Page 10. We notice that our method can have a better clean performance when no trigger is inserted. To intuitively explain this, since the poisoned samples are around the original decision boundary, they do not severely change the decision boundary, thus preserving a high clean accuracy when no attack is injected in the testing data.
>
> **5. Formal definition of stealthiness**
>
> We thank the reviewer for raising this concern. We include a detailed discussion of stealthiness in **Page 26, Section 7.6**.  In this work, we focus on the ``stealthiness" that poisoned samples are separable in the latent space given the labels. Assume the backdoored model $f$ mapping from input space to the latent representation space, classifier $g$ with label space $Y$, sample $x$ and trigger $t$, and also assume the true label for $x$ as $y$, the triggered label is $y_t$, i.e. $g(f(x))=y$ and $g(f(x+t))=y_t$ where $x+t$ denote the combination of input $x$ and trigger $t$. Then the stealthiness is measured by the distance between poisoned sample $x+t$ and the center of clean samples from the target class $y_t$ in the latent space, i.e.
> $d(f(x+t), f(x)_{y,mean})$
> A larger distance means the poisoned sample is separated from the target class and therefore easy to detect, and vice versa.
> In the proposed method, since we take the model output before the last linear layer as the latent representation (for example ResNet18), selection based on the confidence score is equivalent to that directly based on the latent representation.
>
> To provide some more details, we visualize two classes in Cifar10 (shown in the revised paper, Figure 10 (a,b) Page 26). In these figures, there are two clusters with different colors. The blue one is the target class, the green one is the original class, and the red points are the poisoned samples. These verify our statements. In the random selection case, since the poisoned samples are far away from the blue cluster, with the label as the blue class, they are more likely to be identified by the defender. For boundary selection, the poisoned samples are less likely to be detected.
>
> Besides the formal definition above, we would like to clarify that from a more general perspective, the stealthiness of an attack is to what extent the attack can be defended. Since defenses are based on different insights, for instance, Spectral Signature (SS) is based on outlier-detection while anti-backdoor (ABL) is based on the fact that poisoned sample is learned faster than clean data, it is hard to find a formal definition for stealthiness accommodating all defenses. In our experiments, we test different defenses and the reduction of success rate measure the ``stealthiness" (smaller reduction means better resistance against defenses thus more stealthiness).

---

> ### Author Response · Authors · 2024-09-25
> **Response to Reviewer RvbD 3/**
>
> **6. Clarification of Fugures 2a, 2b**
>
> We thank the reviewer for raising this comment. The distances in the figures are indeed non-negative. We present the kernel density curve rather than the histogram of the original data to better compare the distribution of distances. This is why there are negative regions. We have included the histogram of original data and it is clear that all distances are non-negative. We clarify it in the revision, **Page 25, Section 7.5**.
>
> **7. Clarification of image generator $G_t$**
>
> We thank the reviewer for pointing this out. The generator $G_t$ is a transformation: $\mathcal{X}\rightarrow \mathcal{X}$ and satisfies some constraints: $G_t(x)\ne x$ and $d(G_t(x),x)\le \epsilon$ where $d$ is some distance function such as $l_2/l_{\infty}$ distance. The first constraint means that the trigger $t$ must be incorporated, and the second constraint means that the difference between the perturbed image and the original one should be imperceptible. This is included in revision, **Page 3** after equation (1).
>
> **8. Formal definition of distances $d_o,d_t$**
>
> We provide a formal definition for both distances. Assume sample $x$ comes from a class $y$ and the target class is $y_t$, model $f$ mapping from input space to the latent space. Let $f(x)_{y,mean}$ denote the center of samples in class $y$ in the representation space of clean model $f$ and
>
> $\widetilde{f}(x)_{y_t,mean}$ denote the center of samples in class $y_t$ in the representation space of backdoored model $\widetilde{f}$.
>
> Then we can define $d_o, d_t$ as:
>
> $
> d_o(x):=\|f(x)-f(x)_{y,mean}\|_2
> $
>
> $d_t(x):=\|\widetilde{f}(x+t)-\widetilde{f}(x)_{y_t,mean}\|_2$
>
> This formal definition is included in **Page 25, Section 7.5** and we also mention it in the footnote of **Page 4**.
>
> **9. Minor Editorial Clarifications**
>
> We have fixed the typos and made our presentation clearer. Please check the highlighted sentences on **Page 1, 2, 3, 4**.

---

> ### Author Response · Authors · 2024-09-27
> **A friendly reminder**
>
> We are grateful for your comments. We carefully revise the paper and provide detailed responses to address all your concerns. Please let us know if you have further concerns. We are looking forward to more discussion.

---

### Review · Reviewer_nHsq · 2024-09-13

**Summary Of Contributions:**

This work studies the problem of crafting a backdoor trigger attack that can bypass existing methods for defending against backdoor attacks. The method is motivated by the observation trigger attacks produce samples that are far from the typical sample in the classifier latent space, which can be easily detected by defenders. To overcome this, the work proposes to replace random selection of samples for poisoning to selection of samples that are close to the decision boundary for a given poison target class (as selected by a surrogate model available to the attacker). Experiments show that the method can improve attack effectiveness against SOTA defense methods, at the cost of attack effectiveness for an undefended model.

**Audience:**

Yes

**Broader Impact Concerns:**

Not discussed, but this does not impact my assessment.

**Claims And Evidence:**

Yes

**Requested Changes:**

My main requested change is a clear discussion of the reduced effectiveness of the proposed method compared to other attacks on undefended models, and an explanation for this phenomenon.

**Strengths And Weaknesses:**

Strengths:
* The work takes a unique approach to backdoor poisoning based on sample selection for poisoning. The intuitive motivation of using poison samples that are close to the decision boundary, so that poisoned samples are closer to the typical sample in the target poison class, makes sense.
* Experiments show significant improvement in attack effectiveness against representative defense methods.
* The motivating plots in Figures 1 and 2 support the arguments made in the text.
* The paper is well-written and easy to understand.

Weaknesses:
* The primary weakness of the proposed method is the significant decrease in the attack effectiveness against undefended models. This phenomenon is perhaps not highlighted as a limitation strongly enough. A solid explanation for this phenomenon and possible approaches for fixing it would greatly strengthen the paper.
* The techniques used are fairly straightforward. Given the major drawbacks of the proposed method when it comes to attacks against undefended models, I wonder if developing the attack further could address this shortcoming.
* This is not a major weakness, but it is worth wondering if the attack would be successful against a defender who was aware of this specific attack strategy.

---

> ### Author Response · Authors · 2024-09-25
> **Response to Reviewer nHsq**
>
> We thank the reviewer for these valuable comments and we address the concerns respectively.
>
> **1. Discussion of the reduced effectiveness of CBS on undefended models, and what strategy to overcome this limitation.**
>
> We thank the reviewer for raising this comment. As shown in **Theorem 4.2, Theorem 4.3, and Remark 4.8**, we observe an effectiveness-stealthiness trade-off when selecting samples to poison: When the selected samples are closer to the boundary, it is harder to detect them because they are closer to the target class, but the success rate is also reduced because there is only a small change in the decision boundary after poisoning. Our experiments are aligned with this theoretical intuition. We highlight this in the revision, **Page 8 Remark 4.8 and Page 11 Performance comparison in Section 5.2.**
>
> We design two strategies to mitigate this limitation. The general idea is to balance the effectiveness-stealthiness trade-off to control the performance change of different defense methods. In the first strategy, we set a lower threshold for confidence score during selection, i.e. $\epsilon_1 \le |s_c(f(x_i;\theta), y_i)-s_c(f(x_i;\theta), y_t)|\le \epsilon_2$. The second strategy is to mix the boundary samples with some random samples. We conduct experiments to test if these strategies can improve the performance on undefended models. We test with ResNet18 model, Cifar10 dataset, and the backbone attacks are BadNet and Blended. For the first strategy, we set the threshold as [0.07, 0.25] (about 100 samples within this interval and a good balance between effectiveness and stealthiness), and for the second strategy, we select 70\% boundary samples and 30\% random samples. The poison rate is 0.2%, i.e. 100 poisoned samples. We report results of both undefended and 4 representative defenses.
>
> | Cifar10    | ResNet18 | Random | FUS  | CBS  | CBS+1 | CBS+2 |
> |------------|----------|--------|------|------|-------|-------|
> | No defense | BadNet   | 99.9   | 99.9 | 93.6 | 96.3  | 98.5  |
> |            | Blend    | 89.7   | 93.1 | 86.5 | 87.3  | 88.9  |
> | SS         | BadNet   | 0.5    | 4.7  | 20.2 | 18.8  | 17.5  |
> |            | Blend    | 43.7   | 42.6 | 55.7 | 53.2  | 51.8  |
> | STRIP      | BadNet   | 0.5    | 8.5  | 23.7 | 22.3  | 20.9  |
> |            | Blend    | 54.7   | 57.2 | 60.6 | 59.1  | 58.4  |
> | ABL        | BadNet   | 16.8   | 17.3 | 31.3 | 29.6  | 27.7  |
> |            | Blend    | 57.2   | 55.1 | 65.7 | 63.5  | 61.8  |
> | SPECTRE    | BadNet   | 0.9    | 10.1 | 19.6 | 17.5  | 16.4  |
> |            | Blend    | 9.2    | 16.7 | 24.2 | 22.1  | 21.3  |
>
> From the above result, both strategies can improve the attacking performance under no defenses and can achieve comparable performance with random selections. The performance under defenses is slightly reduced because the poisoned samples are easier to detect by defense methods. Nonetheless, these two methods still significantly outperform baselines. We include this discussion in the revision, **Page 28 Appendix 7.8** and mention it in the footnote of **Page 11** in the main content.
>
> **2.** Our discussion in 1 shows that some techniques can be applied to improve the effectiveness and reduce the shortcomings.
>
> **3. Discussion on whether the proposed attack is still successful when known by defenders.**
>
> We agree that when the defender is aware of CBS, it is possible to design ways to identify poisoned samples. For example, the defender can utilize a surrogate model to check the confidence of the samples, mitigating the poisoning effect. However, since the samples around the decision boundary are crucial in guiding model training [1], it is hard to perfectly defend against CBS while maintaining effective model training. In addition, the attacker can also conduct adaptive attacks if the defender's strategy is known, e.g., replacing CBS with CBS+1 or CBS+2.
>
>
> **References**
>
> [1] Not all samples are created equal: Deep learning with importance sampling. A Katharopoulos et al, 2018

---

> ### Author Response · Authors · 2024-09-27
> **A friendly reminder**
>
> We are grateful for your comments. We carefully revise the paper and provide detailed responses to address all your concerns. Please let us know if you have further concerns. We are looking forward to more discussion.

---

### Author Response · Authors · 2024-09-25
**Summary of revision**

We appreciate the reviewers providing constructive suggestions to improve our paper. In addition to the detailed feedback to the specific comments, below is a summary of significant changes in the revision.

* In Section 4.3, we replace the original Theorem 4.2 with more theorems to include a more detailed analysis. In the revision, Theorem 4.2 provides a finite-sample analysis of the Mahalanobis distance; Theorem 4.3 provides an analysis of success rate given infinite samples for a fixed poisoned sample; Remark 4.4 discusses the effect of triggers and selected poisoned samples to the margin of SVM; Theorem 4.5 provides a finite-sample analysis of the margin, with further discussions in Proposition 4.6 and Proposition 4.7 for the proposed CBS method and random selection.

* A variety of discussions and new experimental results are provided to discuss clean accuracy, additional datasets, effectiveness-stealthiness trade-off, the effect of poisoning rate, the effect of surrogate models, the relationship between raw input space and latent space. While the detailed results are included the Appendix (mainly in Section 7), we mention the results in the main paper as well.

All the revised materials are updated in the PDF, and we use blue color to highlight the changes.

---

### Decision · Action_Editor_zAy4 · 2024-11-07

**Recommendation:** Accept as is

**Comment:**

The paper introduces a novel and intuitive method for selecting examples for backdoor attack based on their distance to the decision boundary. This improves success rates across a range of datasets and attack methods. The empirical results are further supported by theoretical analysis.

Reviewers were all supportive of accepting the paper and appreciated the clarity of the paper.

Reviewers have raised various concerns that from my understanding were largely addressed. In particular, Reviewer RvbD raised the concern that the threat model is weaker than a reader might understand, as it requires access to the full training dataset. Authors have addressed this comment in the revised version.

All in all, it is my pleasure to recommend accepting the paper.

**Audience:**

The paper will be of interest to the community focused on AI threats, especially attacks on the training set.

**Claims And Evidence:**

The main claim is that selecting examples close to the decision boundary results in a harder to detect backdoor attack. The claim is backed by qualitative, quantitative and theoretical results.